# Nuclear genome of *Bulinus truncatus*, an intermediate host of the carcinogenic human blood fluke *Schistosoma haematobium*

Neil D. Young [1✉], Andreas J. Stroehlein[1], Tao Wang[1], Pasi K. Korhonen [1], Margaret Mentink-Kane[2], J. Russell Stothard [3], David Rollinson [4,5] & Robin B. Gasser [1✉]

Some snails act as intermediate hosts (vectors) for parasitic flatworms (flukes) that cause neglected tropical diseases, such as schistosomiases. *Schistosoma haematobium* is a blood fluke that causes urogenital schistosomiasis and induces bladder cancer and increased risk of HIV infection. Understanding the molecular biology of the snail and its relationship with the parasite could guide development of an intervention approach that interrupts transmission. Here, we define the genome for a key intermediate host of *S. haematobium*—called *Bulinus truncatus*—and explore protein groups inferred to play an integral role in the snail's biology and its relationship with the schistosome parasite. *Bu. truncatus* shared many orthologous protein groups with *Biomphalaria glabrata*—the key snail vector for *S. mansoni* which causes hepatointestinal schistosomiasis in people. Conspicuous were expansions in signalling and membrane trafficking proteins, peptidases and their inhibitors as well as gene families linked to immune response regulation, such as a large repertoire of lectin-like molecules. This work provides a sound basis for further studies of snail-parasite interactions in the search for targets to block schistosomiasis transmission.

[1] Faculty of Veterinary and Agricultural Sciences, The University of Melbourne, Parkville, VIC, Australia. [2] NIH-NIAID Schistosomiasis Resource Center, Biomedical Research Institute (BRI), Rockville, MD, USA. [3] Department of Parasitology, Liverpool School of Tropical Medicine, Liverpool, UK. [4] Department of Life Sciences, Natural History Museum, London, UK. [5] London Centre for Neglected Tropical Disease Research, London, UK. ✉email: nyoung@unimelb.edu.au; robinbg@unimelb.edu.au

The phylum Mollusca (molluscs) is represented by at least 70,000 species (55 families), which are essential invertebrates of terrestrial or marine ecosystems[1]. Many act as intermediate hosts (i.e., vectors) of parasites of vertebrates including humans[2]. Key representatives of the latter group (intermediate hosts) are aquatic snails that transmit parasitic trematodes (flukes) which cause some of the most chronic and destructive neglected tropical diseases (NTDs) of humans, including clonorchiasis, opisthorchiasis and schistosomiasis[3,4]. These NTDs (called trematodiases) affect ~280 million people worldwide[3,4] and often cause serious complications, particularly malignant cancers, in chronically affected people[5,6].

Although the control of such trematodiases presently relies heavily on the chemotherapeutic treatment of people with a single drug, called praziquantel[7], the key to effective and sustainable control is to prevent or block disease transmission. One possible way of achieving this is to break the transmission cycle at the level of entry or exit of the parasite from the snail intermediate host. Thus, understanding the fundamental biology of the intermediate host and its relationship with the parasite, particularly at the molecular level, could assist significantly in finding new methods to interrupt transmission and prevent disease.

While there have been major advances in our understanding of the biology and molecular biology of the parasitic trematodes that cause clonorchiasis, opisthorchiasis and schistosomiasis[8–11], this is not the case for the snail intermediate hosts that transmit these diseases. Indeed, the only studies conducted to date[12,13] have reported draft genomes for *Biomphalaria glabrata* – a key intermediate host of the blood fluke *Schistosoma mansoni*, which causes hepato-intestinal schistosomiasis in humans, to underpin studies of snail-schistosome interactions[12,14–18]. However, curated genomic resources are lacking for the vast majority of trematode-transmitting snails[19], which represents a major blind spot for health-related research.

As some of our work over the years has focused on *Schistosoma haematobium*—the causative agent of urogenital schistosomiasis, affecting 100 million people worldwide[20]—we have become particularly keen to create a multi-omic tool box for studying the biology of the key snail intermediate host—*Bulinus truncatus*—of this carcinogenic blood fluke and the molecular interplay between this snail host and the larval stages of the fluke that asexually replicate within it[21].

Recently, we defined a mitochondrial genome that represents an established laboratory line (designated BRI)[22] and a transcriptome of the adult stage of this line of *Bu. truncatus*[23]. In the present study, we logically extend this work, and build on recent success in the assembly of relatively large genomes (400–550 Mb) of invertebrates using third-generation (i.e. long-read or long-range) technology[11,24], to define a reference genome for *Bu. truncatus* using this technology, combined with second-generation (short-read) sequencing and advanced bioinformatics. This work will underpin fundamental studies of the biology, ecology and population genetics of *Bu. truncatus* and the interactions between this snail and *S. haematobium*, with broader implications for comparative genomic and molecular investigations of other snail/pathogen systems.

## Results

**Genome assembly**. From a total of 128.5 Gb (~100-fold coverage) of Illumina short-read, 11 Gb of Oxford Nanopore (~10-fold coverage) and 48 Gb of Hi-C sequence data (Supplementary Table 1), a draft genome (designated Btru.v1) was assembled for *Bu. truncatus*; it contains 11,176 contigs and 523 scaffolds and has a total length of ~1.2 Gb (N50 = 5 Mb; L50 = 68; longest contig = 36.5 Mb) and a GC content of 36.3% (Table 1). Scaffolding with Hi-C data gave ~8.3 million contacts (Supplementary Fig. 1 and Supplementary Table 2), ~7.3 and 1 million of which were within and between chromosomes, respectively; the ~7 million intra-chromosomal contacts were <20 kb in length. In total, 914 of 954 (95.8%) metazoan BUSCO genes were identified in this genome, indicating that the assembly represents a substantial proportion of the complete genome (Table 1).

Subsequently, ploidy was assessed using short read data representing an individual adult specimen of *Bu. truncatus* (Supplementary Fig. 2). Using 21-mers, coverage-peaks were inferred using a diploid (Supplementary Fig. 2a) or tetraploid (Supplementary Fig. 2b) model. Although these models were not unequivocally supported by k-mer frequency coverage or read-mapping to the Btru.v1 assembly (Supplementary Fig. 2c), the "smudgeplot" did infer diploidy (A/B) with minor triploidy (AAB), with a high probability (91%) that the data matched a diploid model and a genome size estimate of ~1.2 Gb (Supplementary Fig. 2d).

**Genome annotation**. A large portion (51.0%) of the genome (Btru.v1) of *Bu. truncatus* was repetitive (Supplementary Table 3) and included transposable elements (~23.7% DNA transposons and 6.1% retrotransposons). Although most DNA transposons could not be classified, hobo-activator hAT transposon-like elements were highly represented (11.4% of the assembly). Retrotransposons were mainly long terminal repeats (LTR; 4.8%) and long interspersed nuclear elements (LINEs; 1.4%). The remaining repeat content included unclassified (18.0%) or simple (2.9%) repeat elements (Supplementary Table 3). The distribution of total repetitive elements across the Btru.v1 genome was relatively even (Supplementary Table 3 and Supplementary Fig. 3). In each 500 kb genomic region of the *Bu. truncatus* genome, LTRs, LINEs, and DNA transposons were encoded on average 47.8, 16.5, 375.8 times, respectively, with no clear association with gene model density (Supplementary Table 3 and Supplementary Fig. 3). The highest frequency of encoded LTRs was observed in scaffolds: HiC_scaffold_34 (n = 174 elements in 500 kb); HiC_scaffold_309 (n = 163); and HiC_scaffold_10 (n = 153). The highest frequency of encoded LINEs was observed in scaffolds:

**Table 1 Features of the genome (Btru.v1) of *Bulinus truncatus*.**

| Assembly | Btru.v1 |
|---|---|
| Number of scaffolds | 523 |
| Total size of scaffolds | 1,221,777,273 |
| Longest scaffold; shortest scaffold | 36,501,513; 10,033 |
| Number of scaffolds of >100 kb; 1 Mb; 10 Mb | 484; 271; 20 |
| N50 scaffold length; L50 scaffold count | 4,956,851; 68 |
| Scaffold %GC | 36.27 |
| Scaffold %N | 0.44 |
| Number of contigs | 11,176 |
| Longest contig | 1,917,814 |
| Number of contigs of >100 kb; 1 Mb | 3717;28 |
| N50 contig length; L50 contig count | 234,265;1488 |
| Contig %GC | 36.43 |
| Genome completeness and accuracy: | |
| Complete BUSCOs[a] | 914 (95.8%) |
| Complete single-copy BUSCOs | 822 (86.2%) |
| Complete and duplicated BUSCOs | 92 (9.6%) |
| Fragmented BUSCOs | 8 (0.8%) |
| Missing BUSCOs | 32 (3.4%) |

[a]Number of Benchmarking Universal Single-Copy Orthologs (BUSCOs) identified (genome-mode) and percentage of the 954 genes within the metazoa_odb10 dataset.

**Table 2 Comparison of the features of the draft genome of *Bulinus truncatus* (Btru.v1) with that of *Biomphalaria glabrata* GCF_000457365.1_ASM45736v1[12].**

| Features | Bulinus truncatus (BRI strain; Btru.v1) | Biomphalaria glabrata (BB02 strain; ASM45736v1) |
|---|---|---|
| Number of genes/mRNA | 26,292/ 26,292 | 25,539, 36,662 |
| Gene length[a] | 11,860 ± 10,842 | 12,166 ± 17061 |
| mRNA length | 1600 ± 1527 | 1925 ± 1795 |
| Coding domain length | 1600 ± 1527 | 1296 ± 1356 |
| Number of exons | 9 ± 12 | 7 ± 8 |
| Exon length | 175 ± 298 | 263 ± 494 |
| Intron length | 1256 ± 2314 | 1603 ± 3933 |
| Protein length | 532 ± 509 | 431 ± 452 |
| Genes with transcriptional support | 19,274 (73.3%) | |
| Completeness: | | |
| Complete BUSCOs[b] | 905 (95.0%) | 847 (88.8%) |
| Complete single-copy BUSCOs | 737 (77.3%) | 825 (86.5%) |
| Complete and duplicated BUSCOs | 168 (17.6%) | 22 (2.3%) |
| Fragmented BUSCOs | 26 (2.7%) | 62 (6.5%) |
| Missing BUSCOs | 23 (2.4%) | 45 (4.7) |

[a]Lengths (bp); mean ± standard deviation.
[b]Number (%) of Benchmarking Universal Single-Copy Orthologs (BUSCOs) identified (in protein mode) using the Metazoa_odb10 dataset (954 genes) for comparison.

HiC_scaffold_188 (*n* = 157); HiC_scaffold_52 (*n* = 140); and HiC_scaffold_188 (*n* = 113). The highest frequency of encoded DNA transposons was observed in scaffolds: HiC_scaffold_252 (*n* = 1062); HiC_scaffold_26 (*n* = 897); and HiC_scaffold_58 (*n* = 773). An analysis of repeat elements (*n* = 25,778) in the proximity (within 5000 nt; both directions) of a protein-coding gene (Supplementary Fig. 3 and Supplementary Table 4) showed that simple (8072 of 519,680 elements) and low complexity repeats (1509 of 65,058) were more likely (*p*-value < 0.01), and DNA transposons (6821 of 983,913) and LTRs (652 of 127,776) less likely (*p*-value < 0.01), to be associated with such genes than with other parts of the genome.

Transcriptomic data for adult *Bu. truncatus* and *Bi. glabrata*, and protein data in the UniProtKB/SwissProt database (14 May 2020)[25], were used for the evidence-based prediction of protein-coding genes from the Btru.v1 genome. A total of 75,434 genes were predicted in the masked assembly, reflecting marked complexity. To enable the characterisation of this gene set, we defined gene clusters (Supplementary Fig. 4a) and selected those in clusters 2 (*n* = 12,209 genes), 4 (*n* = 5238) and 5 (*n* = 9674) as each having salient, common features (i.e. more than one exon, GC content and complexity within predicted protein) (Supplementary Fig. 4b). Thus, a total of 26,292 protein-encoding genes were characterised for *Bu. truncatus* (Table 2) and annotated (Table 3).

The statistics for the annotated genes in the Btru.v1 genome were similar to those of *Bi. glabrata* (BB02 strain)[12]: mean gene lengths (11,860 vs. 12,166 bp for Btru.v1 vs. *Bi. glabrata*), mRNAs (1600 vs. 1925 bp), exons (175 vs. 263 bp) and introns (1256 vs. 1603 bp); mean protein length (532 vs. 431 amino acid [aa] residues) was slightly less than predicted for *Bi. glabrata* (Table 2; Supplementary Fig. 5). Coding regions (1600 *vs.* 1296 bp) and inferred proteins (532 vs. 431 aa residues) were only slightly longer than predicted for *Bi. glabrata* (Table 2; Supplementary Fig. 5). The transcriptome of adult *Bu. truncatus* provided support for 19,274 genes (73.3%; transcripts per million, TPM ≥ 0.2) using RNA short-read (16,938; 64.4%) or long-read (17,137;

65.2%) RNA sequence data sets (Supplementary Fig. 6; Table 2; Supplementary Data 1), with 14,801 genes having transcriptional support using both data sets and displaying a direct association between RNA short-read and long-read TPM values (adjusted R-squared: 0.6589; *p*-value: < 0.01; Supplementary Fig. 6). Long-read data had a mean TPM value of 35.4 and a high coverage (>80%) for 16,216 genes, compared with a TPM value of 28.1 and a high coverage (>80%) for 2507 genes for short-read data.

Within the predicted gene set, we identified 905 (95%) of 954 complete, conserved metazoan genes by BUSCO analysis, suggesting that this set represents most of the genome. This result is similar to that (89%) inferred for complete BUSCOs in *Bi. glabrata* (BB02 strain); however, the Btru.v1 genome was predicted to have more BUSCO gene duplicates (17.6%), but fewer fragmented (2.7%) or missing (2.4%) genes (Table 2). For *Bu. truncatus*, we inferred 23,248 (88.4%) or 18,139 (69.0%) genes encoding proteins with sequence homology to proteins present in the non-redundant UniProt TrEMBL or SwissProt databases, respectively (Table 3).

Subsequently, we functionally annotated 21,951 genes using information from one or more of the following databases: InterProScan (*n* = 20,943; 79.7%), eggNOG (*n* = 20,436; 77.7%) and MEROPS (*n* = 1299; 4.9%) (Table 3; Supplementary Data 1). Gene Ontology (GO) terms were assigned to 18,507 sequences (InterProScan domain: 15,606; eggNOG: 13,009). Most of the encoded proteins (*n* = 26,292) had homology to those in the KEGG database (*n* = 20,433; 77.7%), 15,844 of which were assigned an orthology term (Table 3; Supplementary Data 1 and 2) and represented 45 protein groups or families (KEGG BRITE; Supplementary Data 1 and 2), with most inferred to be associated with "membrane trafficking" (*n* = 1799), "chromosome and associated proteins" (*n* = 1327), "exosome" (*n* = 1220), "peptidases and their inhibitors" (*n* = 1166), "transport system" (*n* = 1124), "ubiquitin system" (*n* = 878) and "G protein-coupled receptors" (GPCRs; *n* = 709). More than half of the genes (*n* = 9494) assigned KO terms were linked to 315 distinct KEGG pathway modules (Supplementary Data 2), including "environmental information processing" – including *signalling*: PI3K-Akt (*n* = 416), calcium (*n* = 309) and cAMP (*n* = 309); neuroactive ligand-receptor interaction (*n* = 679); *cellular community*: focal adhesion (*n* = 433); and *cellular processes*: lysosome transport and catabolism (*n* = 310). Overall, 21,951 (83.5%) of proteins were annotated, and 4341 (16.5%) were not and were, thus, called "orphan" (unknown) proteins (Table 2; Supplementary Data 1). Of these orphans, 2053 had transcriptional support, of which 1549 had homology to proteins predicted for *Bi. glabrata* (BB02 strain).

Excretory/secretory (ES) proteins are reported to play central roles in snail-schistosome interactions[18]. Within the *Bu. truncatus* gene set, 1096 (4.2%) genes were predicted to encode extracellular ES proteins, based on the presence of a signal peptide domain (for 3121 genes; 11.9%) and absence of one or more transmembrane domains (for 18,507 genes) (Table 3; Supplementary Data 1). The full complement of ES proteins (i.e. the secretome) was inferred to represent 434 peptides with KEGG annotations that could be assigned to the following protein groups: peptidases and their inhibitors (*n* = 102), exosome (67), membrane trafficking (48), glycosaminoglycan-binding proteins (47) and lectins (26). Of these 434 proteins, 330 were orphans, 118 had homology to individual proteins in *Bi. glabrata* (BB02 strain), and 251 had transcriptional support in the adult stage of *Bu. truncatus*.

**Marked synteny in mollusc genomes.** The annotated protein-coding gene set was first used to assess conserved gene order (synteny) in Btru.v1 genomic scaffolds compared to genes

**Table 3 Annotation of the protein-encoding genes (n = 26,292) predicted for *Bulinus truncatus*.**

| Description of approach | Genes (% of total) | Unique annotations |
|---|---|---|
| UniProt TrEMBL database annotation | 23,248 (88.4%) | 16,573 |
| UniProt SwissProt database annotation | 18,139 (69.0%) | 11,450 |
| eggNOG database | 20,436 (77.7%) | 12,788 |
| GeneOntology—eggNOG | 13,009 (49.5%) | 7418 |
| Enzymes—eggnog | 5256 (20.0%) | 1129 |
| InterProScan database annotation | 20,943 (79.7%) | 10,618 |
| Pfam annotation | 17,568 (66.8%) | 4933 |
| Gene Ontology | 15,606 (59.4%) | 2755 |
| MEROPS protease database | 1299 (4.9%) | 537 |
| KEGG Orthology (KO) annotation | | |
| KEGG protein families | 15,844 (60.3%) | 6403 |
| KEGG pathways | 9494 (36.1%) | 3464 |
| Annotated by ≥1 method/database[a] | 21,951 (83.5%) | |
| Orphan genes/with transcriptional support | 4341/2053 | |
| Proteins predicted to encode a signal peptide domain | 3121 (11.9%) | |
| Proteins predicted to encode one or more transmembrane domains | 7786 (29.6%) | |
| Proteins predicted to be extracellular/secreted | 1096 (4.2%) | |

[a]Without homology-based matches in the UniProt TrEMBL database (accessed 20 December 2020).

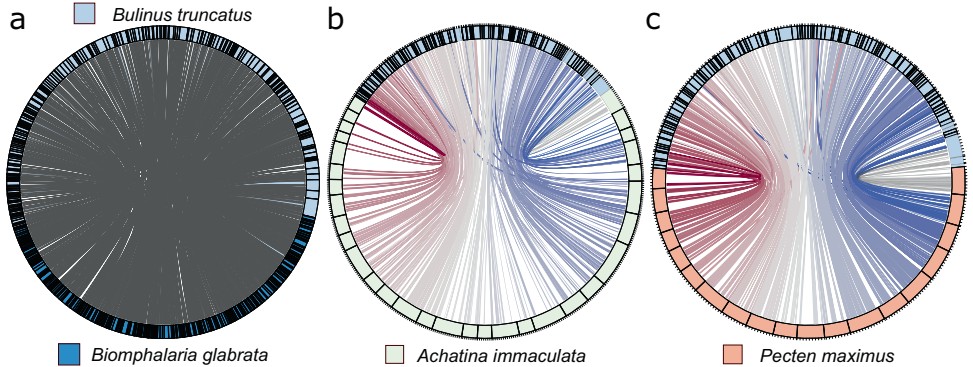

**Fig. 1 Synteny between genomes.** Synteny and contiguity of the genome (Btru.v1) of *Bulinus truncatus* with the draft genome of *Biomphalaria glabrata* and the chromosomal-level reference genomes of each *Achatina immaculata* and *Pecten maximus*, respectively. Scaffolds are arranged in circular (circos) plots with reference scaffolds for *Ac. immaculata* or *P. maximus* linked to inferred syntenic blocks of the *Bu. truncatus* genome using distinctly-coloured bars. Linked syntenic blocks between scaffolds of *Bu. truncatus* and *Bi. glabrata* are shown (light blue). **a** Synteny between the reference genomes of *Bu. truncatus* and *Bi. glabrata* established based on the positions of 595 syntenic blocks each containing three or more single-copy orthologs (i.e. 4047 of a total of 11,051). **b** Synteny between the reference genomes of *Bu. truncatus* and *Ac. immaculata*, established based on the positions of mapped *Bu. truncatus* proteins in 243 syntenic blocks each containing six or more single-copy orthologs (i.e. 2234 of a total of 8517). **c** Synteny between the reference genome of *Bu. truncatus* and *P. maximus* established based on the position of 291 syntenic blocks containing three or more single-copy orthologs (i.e. 1139 of a total of 4685).

encoded in the reference genomes for *Bi. glabrata*, *Achatina immaculata* (gastropod; family Achatinidae) and *Pecten maximus* (bivalve; family Pectinidae); the relationship among these snails is summarised in phylogenetic analyses of concatenated amino acid sequence data inferred from 2315 SCOs using maximum likelihood (ML) and Bayesian inference (BI) methods (Supplementary Fig. 7). Upon pairwise comparison, many blocks of nucleotide sequence were aligned across conserved gene regions among *Bu. truncatus, Bi. glabrata* (S1316-R1 strain), *Ac. immaculata* and *P. maximus* (Fig. 1), with most conservation seen between Btru.v1 and *Bi. glabrata* (Fig. 2a; Table 4). Almost all genome scaffolds contained regions that aligned in 594 syntenic blocks of ≥3 single-copy orthologs spanning 557.8 Mb (46%) and 396.5 Mb (49 %) of the *Bu. truncatus* (Btru.v1) and *Bi. glabrata* genomes, respectively. For *Bu. truncatus*, most scaffolds (n = 436) had shared synteny with *Ac. immaculata* (Fig. 1b; Table 4), with 97 % (n = 436; ~1.19 Gb) of scaffolds for Btru.v1 aligning to all chromosomes of *Ac. immaculata* (n = 32; 1.65 Gb) in 243 syntenic blocks of ≥6 single-copy orthologs spanning 411.7 Mb (35%)

and 372.6 Mb (23%) of the Btru.v1 and *Ac. immaculata* genomes, respectively. Fewer Btru.v1 scaffolds (n = 398) shared syntenic regions with chromosomes of *P. maximus*, (Fig. 1b; Table 4), with 95 % (n = 398; ~1.17 Gb) of these scaffolds aligning to 19 of the 3983 scaffolds (92%; 844.3 Mb) in 291 syntenic blocks of ≥3 single-copy orthologs spanning 404.9 Mb (35%) and 534.2 Mb (63%) of the Btru.v1 and *P. maximus* genomes, respectively.

**Protein ortho-groups and expansions.** Next, the annotated *Bu. truncatus* protein-coding gene set was compared to genome annotations available for *Bi. glabrata* (BB02 strain), *Aplysia californica*, *Elysia chlorotica* and *P. maximus* (Table 5 and Fig. 2); the relationship among these snails is summarised in phylogenetic analyses of concatenated amino acid sequence data inferred from 2315 SCOs using maximum likelihood (ML) and Bayesian inference (BI) methods (Supplementary Fig. 7) Upon comparison, 17,866 ortho-groups containing one or more proteins were identified. Most ortho-groups (n = 13,302) were shared by *Bu. truncatus* (n = 20,349 proteins) and *Bi. glabrata* (n = 19,853 proteins);

**Table 4 Genome-wide synteny comparisons of the genomes of *Bulinus truncatus* (Btru.v1), *Biomphalaria glabrata* (1316-R1/ASM1452496v1), *Achatina immaculata* and *Pecten maximus* in a pairwise manner.**

| Comparison | Species | Genome size | Total no. of scaffolds | Scaffolds aligned (%) | Total no. of scaffolds aligned | Syntenic blocks/ length (%) | No. of single-copy orthologs in syntenic blocks | No. of single-copy orthologs |
|---|---|---|---|---|---|---|---|---|
| *Bu. truncatus* vs. *Bi. glabrata* | *Bu. truncatus* | 1.23 Gb | 523 | 1.21 Gb (99%) | 472 | 594/ 557.8 Mb (46%) | 6939 | 11,051 |
| | *Bi. glabrata* | 852.0 Mb | 927 | 807.7 Mb (95%) | 526 | 594/ 396.5 Mb (49%) | 6939 | 11,051 |
| *Bu. truncatus* vs. *Ac. immaculata* | *Bu. truncatus* | 1.23 Gb | 523 | 1.19 Gb (97%) | 436 | 243/ 411.7 Mb (35%) | 2234 | 8517 |
| | *Ac. immaculata* | 1.65 Gb | 31 | 1.65 Gb (100%) | 31 | 243/ 372.6 Mb (23%) | 2234 | 8517 |
| *Bu. truncatus* vs. *P. maximus* | *Bu. truncatus* | 1.23 Gb | 523 | 1.17 Gb (95%) | 398 | 29/ 404.9 Mb (35%) | 1139 | 4685 |
| | *P. maximus* | 918.3 Mb | 3983 | 844.3 Mb (92%) | 19 | 29/ 534.2 Mb (63%) | 1139 | 4685 |

4583 *Bu. truncatus* ortho-groups (n = 5943 proteins) and 4252 *Bi. glabrata* ortho-groups (n = 5685 proteins) were unique to each of these two species (Fig. 2). In total, 7895 ortho-groups containing one or more proteins were shared by *Bu. truncatus*, *Bi. glabrata*, *Ap. californica*, *E. chlorotica* and *P. maximus* (Fig. 2), with 3924 ortho-groups (n = 5079 proteins) being unique to *Bu. truncatus* and 1695 ortho-groups representing both *Bu. truncatus* (n = 2436 proteins) and *Bi. glabrata* (n = 2999 proteins), to the exclusion of the other mollusc species studied (Fig. 2c). Of all ortho-groups shared by all species, 2919 were single-copy orthologs (Fig. 2d). Maximum likelihood (ML) and BI analyses of aligned single-copy gene sequence data placed *Bu. truncatus* with *Bi. glabrata*, and *Ap. californica* with *E. chlorotica*, with absolute nodal support (Fig. 2d).

Subsequently, we explored protein family expansions in *Bu. truncatus*, or in *Bu. truncatus* and *Bi. glabrata* (BB02 strain) (Fig. 2e) with respect to other mollusc species included. Of the 5079 proteins (3924 ortho-groups) unique to *Bu. truncatus*, 1898 were classified into one of 31 (KEGG BRITE) families encoded by at least 10 genes (Fig. 2; Supplementary Data 1). Of the 2436 proteins in *Bu. truncatus* with an orthologue exclusively in *Bi. glabrata*, 1039 were classified into 20 families encoded by at least 10 genes (Fig. 2; Supplementary Data 1). Protein families unique to *Bu. truncatus* were associated with *metabolism*: peptidases and inhibitors (n = 185), protein kinases (n = 94), protein phosphatases (n = 68) and glycosyltransferases (n = 62); *genetic information processing*: membrane trafficking (n = 236), the ubiquitin system (n = 110), chromosome (n = 107), DNA repair and recombination proteins (n = 37) and spliceosome (n = 37); and *signalling and cellular processes*: exosome (n = 209), GPCRs (n = 157), glycosaminoglycan binding proteins (n = 111), transport system (n = 110) and cluster of differentiation (CD) molecules (n = 73). Protein families shared by *Bu. truncatus* and *Bi. glabrata* were associated with metabolism: peptidases and their inhibitors (n = 101), protein kinases (n = 61), protein phosphatases (n = 42) and glycosyltransferases (n = 38); genetic information processing: membrane trafficking (n = 126), ubiquitin system (n = 39), chromosome (n = 24) and transcription factors (n = 28); and *signalling and cellular processes*: GPCRs (n = 134), exosome (n = 107), transport system (n = 91), glycosaminoglycan binding proteins (n = 73) and CD molecules (n = 80).

**Protein groups inferred to be involved in the snail-schistosome relationship.** Given the role of *Bu. truncatus* as a key intermediate host for *S. haematobium* and related schistosome taxa[21], we explored protein groups inferred to be involved in the immune/defence system of the snail and/or interactions with schistosomes (Table 6; Fig. 3; Supplementary Table 5).

Proteins of the Guadeloupe resistance complex (GRC)[26] and the polymorphic transmembrane cluster 2 (PTC2)[13], proposed to be loci associated with parasite recognition and/or reduced susceptibility to schistosome infection[13,26], were inferred for *Bu. truncatus*. The 11 GRC homologs predicted (Fig. 3; Table 6) were linked to cellular processes (lysosome and adherens junction) and metabolic processes required for glycosaminoglycan degradation, and five PTC2 homologs (Table 6, Fig. 3 and Supplementary Table 5) linked to metabolic processes required for glycosphingolipid biosynthesis.

The baculovirus inhibitor of apoptosis (IAP) repeat (BIR) protein homologs (n = 117) identified in *Bu. truncatus* were inferred to be involved in apoptosis, ubiquitin-mediated proteolysis, necroptosis and NF-kappa beta-signalling pathways (Table 6). Most BIR proteins (n = 25) of *Bu. truncatus* were represented within ortho-group OG0000004 containing 43, 1, 9

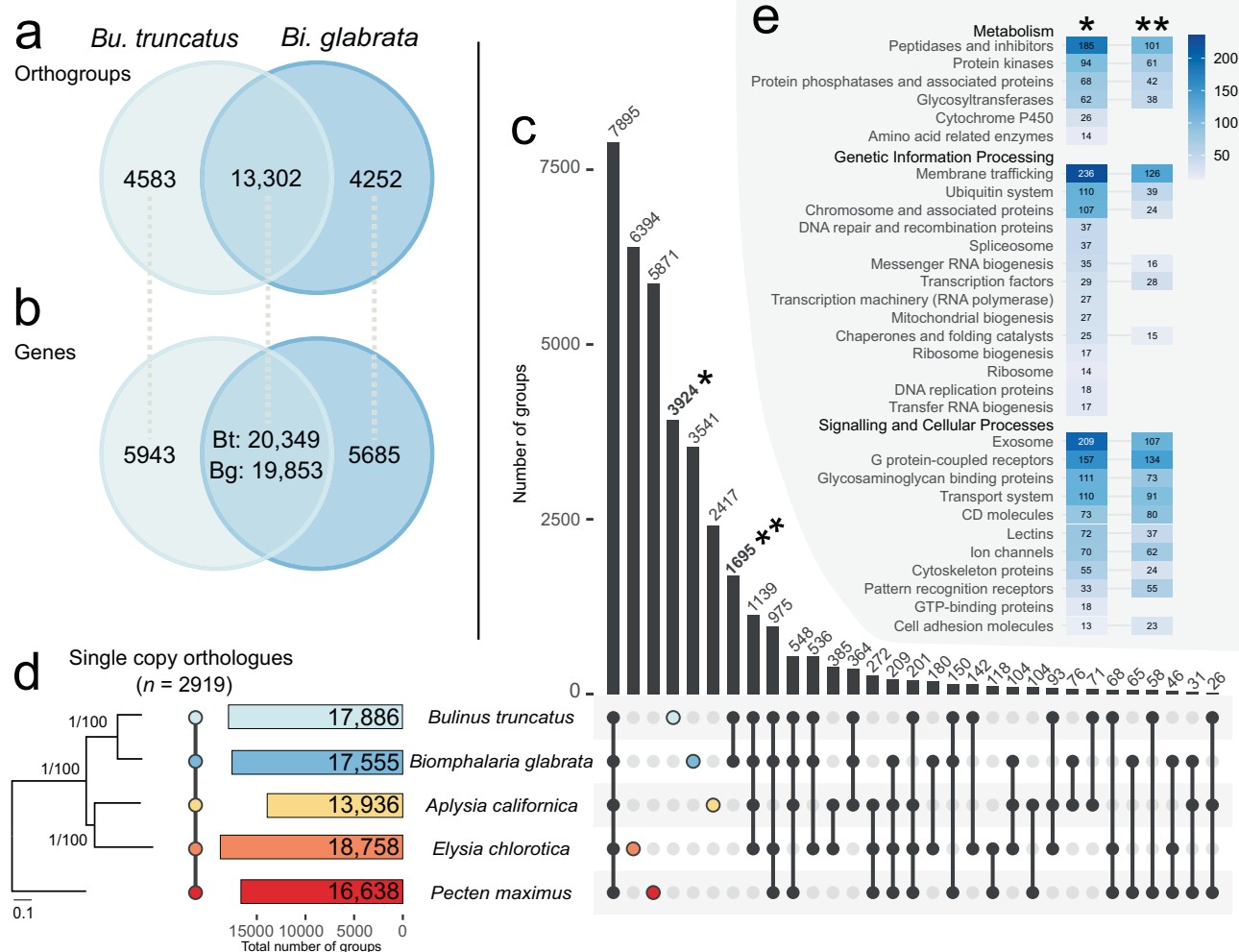

**Fig. 2 Orthologous protein groups.** Orthologous groups of one or more protein(s) in gastropod taxa with available gene annotation. *Pecten maximus* is an outgroup (bivalve). **a** Pairwise comparison of the orthologous groups common to *Bulinus truncatus* and *Biomphalaria glabrata*. **b** Genes in orthologous groups common to *Bu. truncatus* and *Bi. glabrata*. **c** UpSet plot of the intersections of unique or shared orthologous groups inferred from protein data sets for *Bu. truncatus*, *Bi. glabrata*, *Aplysia californica*, *Elysia chlorotica* and *P. maximus*. **d** Phylogenetic tree inferred from single copy orthologs aligned among selected gastropod taxa, including all available annotated genes. As the topology of the maximum likelihood (ML) and Bayesian inference (BI) trees was the same, the ML tree is displayed and shows nodal support values for both BI (pp) and ML (bootstrap). **e** Genes (annotated by KEGG) predicted to be unique to *Bu. truncatus* (coloured), or shared between or among the five molluscan species included here.

and 8 respective orthologues of *Bi. glabrata*, *Ap. californica*, *E. chlorotica* and *P. maximus* (Fig. 3). These findings are consistent with earlier reports showing an expansion of BIR proteins in molluscs[12].

An expanded set of Toll-like receptors ($n = 123$) with a Toll/ interleukin-1 receptor homology domain (IPR000157; Table 6) associated with Toll-like signalling, necroptosis, NF-kappa beta and/or hypoxia-inducible factor 1 (HIF-1) signalling pathways was identified. Most Toll/interleukin-1 proteins ($n = 15$) of *Bu. truncatus* were in ortho-group OG0000039, together with 27 orthologues in *Bi. glabrata* and 8 in *Ap. californica* (Fig. 3).

Likely to be involved in reduced snail susceptibility to schistosome infection[18] are cathepsin homologs ($n = 21$) of *Bu. truncatus*, commonly involved in transport and catabolism (lysosome, phagosome and autophagy), cell growth and death (apoptosis) and antigen-processing/presentation pathways (Table 6). Ten of these cathepsins were within ortho-group OG0000133, together with 9, 5, 1 and 4 respective orthologues of *Bi. glabrata*, *Ap. californica*, *E. chlorotica* and *P. maximus* (Fig. 3). Of the chitinases ($n = 103$) predicted to be involved in amino sugar and nucleotide sugar metabolism in *Bu. truncatus* (Table 6)

and reduced snail susceptibility to schistosome infection, 36 were within ortho-group OG0000085, but had no ortholog in any other mollusc species studied (Fig. 3). In addition, of the calmodulins ($n = 42$) that likely associate with pathogen interactions[27] (Tables 6), 6 were within ortho-group OG0000309, also with 6, 4 and 4 respective orthologues of *Bi. glabrata*, *E. chlorotica* and *P. maximus* (Fig. 3).

Lectins ($n = 101$) identified in *Bu. truncatus* were predominantly assigned to the C-type lectin receptor signalling pathway, being associated with roles in focal adhesion, interactions with the extracellular matrix (ECM) and phagosomes/lysosomes (Table 6 and Fig. 3). For instance, 18 mannose receptor C type-like proteins ($n = 18$) were identified to have 20, 9 and 6 orthologues in *Bi. glabrata* *Ap. californica* and *E. chlorotica*, respectively (Fig. 3). A group of lectin-like proteins, homologous to fibrinogen-related proteins of *Bi. glabrata* (FReD; InterPro identifier IPR036056), were also inferred for *Bu. truncatus* ($n = 130$). The abundance of FReDs ($n = 130$) in *Bu. truncatus* contrasted the smaller number ($n = 72$) in *Bi. glabrata*, and the closest homologs between the two species were in distinct ortho-groups (Table 6 and Fig. 3). Most FReDs of *Bu. truncatus*

**Table 5 Molluscan species/strains studied herein and relevant information on their genomes.**

| Species (strain) | Genome code | Accession number | Gene annotation | Assembly level | Scaffold N50 | Size | Reference |
|---|---|---|---|---|---|---|---|
| *Achatina immaculata* | ASM976088v1 | GCA_009760885.1 | No | Chromosome | 56.4 Mb | 1.65 Gb | Unpublished |
| *Aplysia californica* | AplCal3.0 | GCF_000002075.1 | Yes | Scaffold | 917.5 kb | 927.3 Mb | Unpublished |
| *Biomphalaria glabrata* (BB02) | ASM45736v1 | GCA_000457365.1 | Yes | Scaffold | 48.1 kb | 916.4 Mb | 12 |
| *Biomphalaria glabrata* (1316-R1) | ASM1452496v1 | GCA_014524965.1 | No | Scaffold | 2.60 Mb | 852.0 Mb | 13 |
| *Elysia chlorotica* | ElyChl2.0 | GCA_003991915.1 | Yes | Scaffold | 442.0 kb | 557.5 Mb | 88 |
| *Pecten maximus* | xPecMax1.1 | GCF_902652985.1 | Yes | Chromosome | 44.8 Mb | 918.3 Mb | 89 |

represented fibrinogen C-terminal domain proteins within ortho-group OG0000016 (38 proteins), also containing 8, 14 and 2 orthologs of *Bi. glabrata*, *Ap. californica* and *P. maximus*, respectively (Fig. 3).

**Classification of fibrinogen-related proteins.** The phylogenetic analysis of aligned amino acid sequences of the conserved C-terminal fibrinogen domain (FBG) of FReDs of *Bu. truncatus* ($n = 95$) and *Bi. glabrata* ($n = 56$) inferred 7 groups, of which Groups 1 to 5 and 7 had strong nodal support (pp = 0.93 to 1.0; Fig. 4a). For both snail species, individual proteins in these groups were annotated/classified based on their predicted tertiary structures (Fig. 4b, Supplementary Table 6 and Supplementary Fig. 8; Supplementary Data 3). Class A proteins matched *Bi. glabrata* sFReD proteins with a single C-terminal fibrinogen-like domain (Fig. 4b, Supplementary Fig. 8). Unlike sFReDs, some Class A proteins contained N-terminal α-helices (Class A α) or β-sheets and α-helices (Class A βα). The presence of α-helices in Class A α proteins in close proximity to the N-terminus was suggestive of a signal peptide, as they often preferentially adopt an α-helical form[28]. Class B and C proteins matched *Bi. glabrata* FREP proteins, with a single FReD domain linked to one (Class B) or two (Class C) N-terminal immunoglobulin superfamily (IgSF) domains by α-helices (Supplementary Table 6 and Supplementary Fig. 8). Class D proteins were *Bi. glabrata* FREM-like proteins (Supplementary Table 6). Class E and F were *Bu. truncatus* FReD-like proteins with complex structures in the N-terminus but had no structural homology to known, predicted *Bi. glabrata* proteins (Fig. 4b). Some (Class A-like) proteins contained a partial FBG domain (Supplementary Table 6), and, thus, had been excluded from the original phylogenetic analysis (Fig. 4a).

A detailed exploration of fibrinogen-related proteins in these phylogenetic groups based on their predicted structures provided enhanced insights. Group 1 contained 36 *Bi. glabrata* proteins, 21 of which had a predicted signal peptide based on primary amino acid sequence (Fig. 4a). Group 1 proteins were classified as Class B FREPs, ($n = 9$), Class C FREPs ($n = 20$) or sFReDs with (Class A α $n = 4$; Class A βα $n = 1$) or without (Class A; $n = 1$) additional N-terminal tertiary structures. Group 2 included 27 *Bu. truncatus* proteins, 16 of which had signal peptides. Most Group 2 *Bu. truncatus* proteins were classified as sFReDs without (Class A; $n = 14$) or with additional N-terminal structures (Class A α n = 4; Class A βα $n = 7$). Many proteins in Group 2 were assigned to ortho-group OG0000016, including one *Bu. truncatus* protein (Btru_033719) with a distinct N-terminal domain encoding several, ordered β-sheets (Class E). The three *Bi. glabrata* proteins within Group 2 all encoded a signal peptide and were classified as sFReDs (n = 2) or Class A α ($n = 1$) (Fig. 4a). Group 3 contained only two *Bi. glabrata* proteins and were classified as Class A α. Group 4 included 58 *Bu. truncatus* and 5 *Bi. glabrata* proteins, 33 and 3 of which had signal peptides, respectively. Proteins in Group 4 were most diverse, including FReDs assigned to ortho-groups OG0000103, OG0000581,

OG0010992 and OG0015404. *Bi. glabrata* proteins in Group 4 were all predicted to be sFReDs and represented a distinct subgroup with three *Bu. truncatus* proteins (Fig. 4b; pp = 0.85). Most *Bu. truncatus* in Group 4 were sFReDs with (Class A α $n = 13$; Class A βα $n = 2$) or without additional N-terminal structures (Class A; $n = 37$) or were predicted to encode only a partial FReD domain ($n = 5$). One FReD-like protein (Btru_048110) in this group encoded a novel N-terminal domain (Class F; Fig. 4b), but clustered with *Bu. truncatus* sFReDs (Class A). Group 5 proteins were all classified as Class A α (4 *Bu. truncatus* and 2 *Bi. glabrata*), two of which were assigned to ortho-group OG006406. Only one protein (*Bu. truncatus* Btru_048819) in this group had a predicted signal peptide. Group 6 proteins all had signal peptides; two were *Bi. glabrata* proteins (one sFReD and one in Class C). Interestingly, the tertiary structure of a Class C-like *Bu. truncatus* protein (Btru_007887) in this group encoded two IgSF-like domains (cf. Figure 4b), which had not been detected previously based on amino acid sequence homology (Supplementary Data 1); this *Bu. truncatus* protein had a shorter FReD domain and a longer chain of α-helices than seen for *Bi. glabrata* Class C FREPs (Supplementary Fig. 8). Group 7 contained three *Bu. truncatus* and three *Bi. glabrata* protein, all of which were classified as Class A βα, mostly within ortho-group OG0004042, most of which were not predicted to encode a signal peptide.

## Discussion

Here, we present the first draft nuclear genome (Btru.v1) of a key representative of the snail genus *Bulinus* – a complex of at least 37 species presently divided into four main groups[29]. This draft genome of the BRI-laboratory line of *Bu. truncatus*, originally from Egypt, was assembled using a combination of second- (short-read) and third-generation (long-read) sequence data. The draft assembly shared a high degree of contiguity with the available reference genomes (chromosomes) of the giant land snail, *Ac. immaculata* (Gastropoda: Styllommatophora) and a marine scallop, *P. maximus* (Bivalvia: Pectinida). Most gene models were strongly supported by short- and long-read RNA sequence data, and high-quality models enabled an exploration of the molecular biology of *Bu. truncatus* (BRI strain) and comparative investigations of the genomes of selected molluscs, including *Bi. glabrata*—which is an intermediate host of *S. mansoni*.

*Bu. truncatus* is proposed to have four sets of chromosomes (tetraploid)[29,30], thought to have arisen *via* alloploidy by ancestral hybridisation of closely related diploid species[29]. This proposal is supported by the karyotype and zymograms of an Egyptian isolate of *Bu. truncatus*[31]. However, alternate hypotheses are that this tetraploidy might have resulted from evolutionary saltation(s) upon nuclear fusion of genomes following hybridisation of two distinct diploid species (i.e. allo-tetraploidy) or whole-genome duplication in a diploid ancestor (i.e. auto-tetraploidy). The polymorphism seen in the Btru.v1 genome of *Bu. truncatus*,

**Table 6 Key protein groups in _Bulinus truncatus_ and proposed roles and pathway associations – supported by published information.**

| Protein group | Number of proteins predicted for _Bu. truncatus_ | Number of ortho-groups | Known or proposed roles | Pathway associations | References |
|---|---|---|---|---|---|
| Guadeloupe resistance complex (GRC) | 11 (15)[a] | 8 | Reduces susceptibility to schistosome infection | Cellular processes: lysosome, adherens junction, endocytosis; metabolism: glycosaminoglycan degradation | 26 |
| Polymorphic transmembrane cluster 2 (PTC2) | 8 (11)[a] | 5 | Reduces susceptibility to schistosome infection | Glycan biosynthesis and metabolism; glycosphingolipid biosynthesis | 13 |
| BIRs/IAPs | 117 | 49 | Drug response, apoptosis, innate immune responses | Apoptosis, ubiquitin mediated proteolysis, NF-kappa beta signalling, Toll-like receptor signalling, | 12, 27 |
| Toll-/IL-1-related proteins | 123 | 49 | Reduces susceptibility to schistosome infection, immune response | necroptosis, NF-kappa beta signalling and HIF-1 signaling pathway | 101, 102 |
| Cathepsins | 21 | 7 | Reduces susceptibility to schistosome infection, excretory/secretory product | Transport and catabolism (lysosome, phagosome, autophagy), apoptosis, antigen processing and presentation | 18 |
| Chitinases | 103 | 18 | Reduces susceptibility to schistosome infection, excretory/secretory product | Amino sugar and nucleotide sugar metabolism | 18, 26 |
| Calmodulins | 42 | 28 | Stress response, drug susceptibility | Immune system: C-type lectin receptor signaling pathway | 27 |
| Lectins | 101 | 33 | Immune response | Immune system: C-type lectin receptor signalling pathway | 103 |
| Fibrinogen-related proteins[b] | 130 | 29 | Susceptibility to schistosome infection; immune response | Signalling molecules and interaction, ECM-receptor interaction, focal adhesion | 36, 42–44 |

[a]Number of proteins inferred for _Biomphalaria glabrata_.
[b]A distinct group of lectins with a characteristic C-terminal fibrinogen domain.

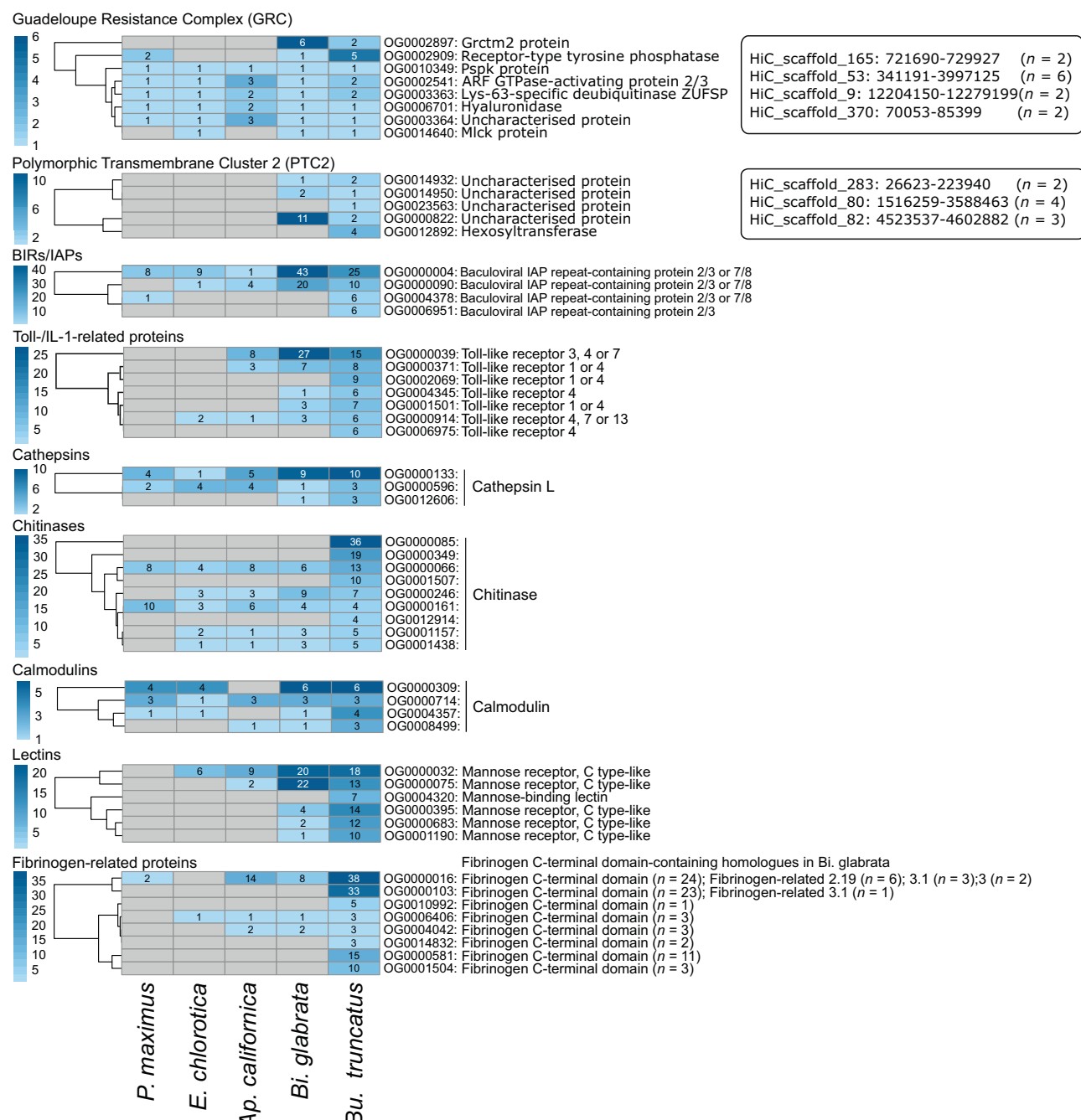

**Fig. 3 Gene expansions in key protein groups.** Expansion of protein ortho-groups in *Bulinus truncatus* predicted to relate to snail-schistosome interactions, based on published information (see Table 6). Cluster dendrograms showing orthogroups for *Bu. truncatus* with the largest gene expansions inferred from protein data sets of *Bu. truncatus*, *Biomphalaria glabrata*, *Aplysia californica*, *Elysia chlorotica* and *Pecten maximus* as defined using OrthoFinder. The locations of genes linked to the Guadeloupe resistance complex (GRC) or polymorphic transmembrane cluster 2 (PTC2) are indicated (boxed).

which seemed to reflect a diploid organism (Supplementary Fig. 2), was consistent with that observed in *Ac. immaculata* which, indeed, underwent whole-genome duplication in a diploid ancestor[32]. This evidence indicates that *Bu. truncatus* is an auto-tetraploid snail, with limited chromosomal divergence. Limited genetic divergence among the four sets of chromosomes and/or the use of a single restriction enzyme (*Dpn*II) for Hi-C library construction, are probable reasons for the *Bu. truncatus* scaffolds being shorter than the expected chromosome lengths. High-coverage long-range and long-read genome sequencing and comparative karyotypic studies of key members of the *Bulinus*

complex, using Btru.v1 as a reference, should establish their chromosomal evolution.

A genome-wide analysis revealed more genomic synteny between *Bu. truncatus* and *Bi. glabrata* 1316-R1 strain[13] (snail hosts of schistosomes) than between *Bu. truncatus* and *Ac. immaculata* (land snail) and *P. maximus* (marine scallop). This finding is consistent with the evolutionary relationships of planorbid/bulinid gastropods[33]. The synteny of large regions (>46% of aligned genomes) of the *Bu. truncatus* and *Bi. glabrata* genomes has loci likely central to the susceptibility of these snails to respective schistosome species[14]. For instance, genomic regions in *Bu. truncatus*

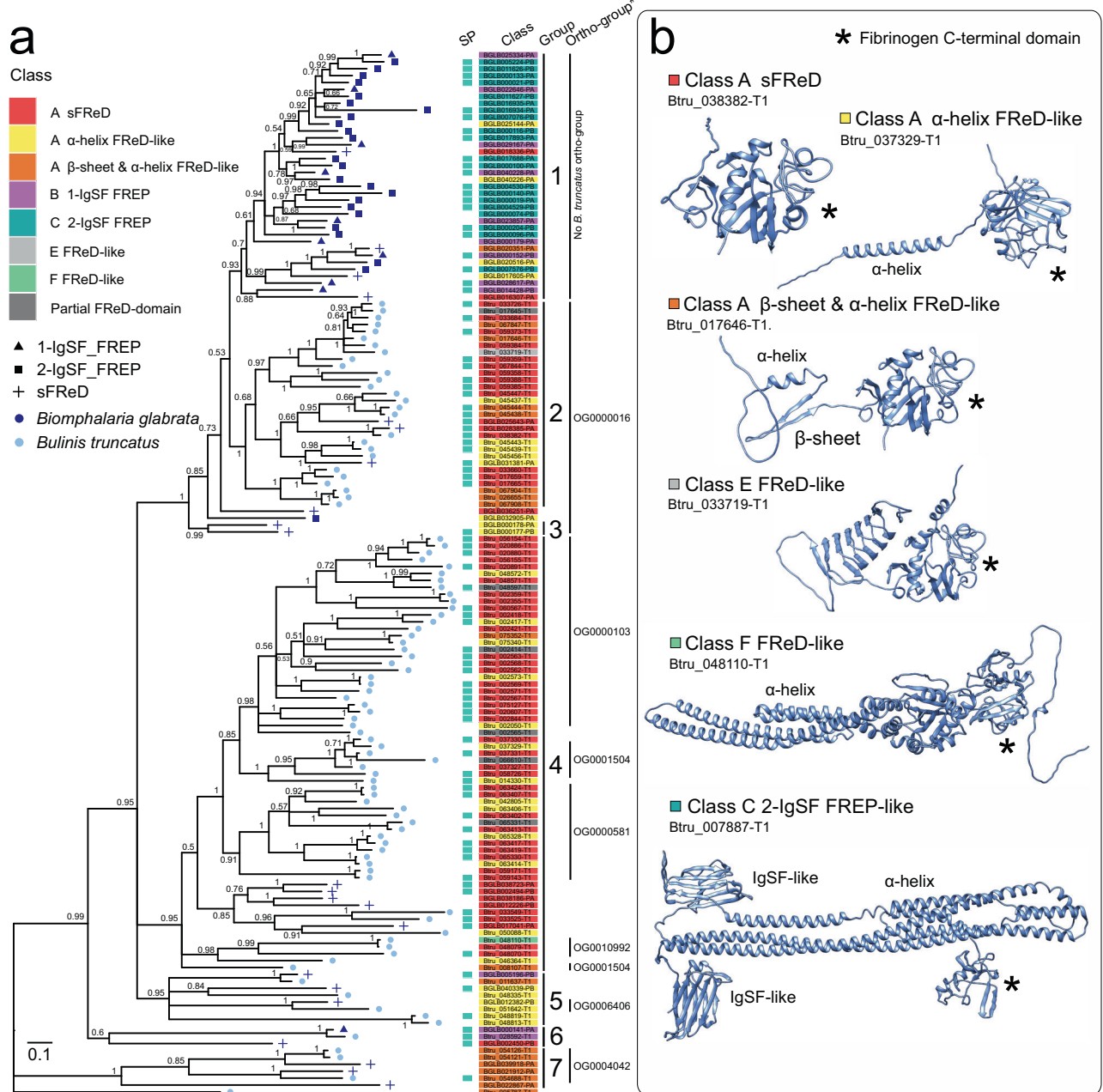

**Fig. 4 Comparison of select fibrinogen-related proteins (FReDs) of *Bulinus truncatus* and *Biomphalaria glabrata*. a** Phylogenetic relationship of a set of curated FReD-like proteins encoded in the genomes of *Bu. truncatus* and *Bi. glabrata*. Each branch tip is labelled with an existing FReD classification system for *Bi. glabrata*[36] (shape) or species (colour). Posterior probability (pp) values are indicated, and groups are numbered (1 to 7). Proteins with predicted signal peptide (SP) domains are indicated with cyan boxes. A distinct colour represents each FReD class, with gene accession numbers boxed. #*Bu. truncatus* ortho-groups inferred from proteins of *Bu. truncatus*, *Bi. glabrata*, *Aplysia californica*, *Elysia chlorotica* and *Pecten maximus* (cf. Figure 3). **b** Tertiary structure models for FReDs of *Bu. truncatus* employed for enhanced classification (A–F). C-terminal domain (*), as well as IgSF-like and α-helix and/or β-sheet structures, are shown.

homologous to loci encoding members of the GRC and PTC2 complexes associated with reduced susceptibility of *Bi. glabrata* to *S. mansoni* infection were identified[13,34]. Detailed comparative studies are now needed to fill the void in our knowledge of the molecular basis of susceptibility in *Bu. truncatus* to *S. haematobium*.

More broadly, a large proportion of proteins inferred for *Bu. truncatus* were orthologous to those predicted for *Bi. glabrata* (BB02 strain) and formed ortho-groups to the exclusion of proteins of the other species of gastropod (*Ap. californica* and *E. chlorotica*) or bivalve (*P. maximus*) studied. These distinct planorbid/bulinid protein groups may be linked to one or more

evolutionary events which led to the speciation within the superfamily Lymnaeoidea. Expansions of some protein-gene families/groups in *Bu. truncatus* were relatively consistent with those in *Bi. glabrata*[12,14,35–37]. For instance, shared expansions were seen for some genes encoding GPCRs, proteins involved in membrane trafficking, and peptidases and their inhibitors. The large repertoire of genes encoding GPCRs and transport proteins likely relates to the diversity of receptors in olfactory organs of gastropods with key chemosensory roles[12,38].

Key expansions of select gene families could have arisen during the evolution of snail defences against pathogens, including

schistosomes. For example, the cathepsin L-like genes of *Bu. truncatus* have homologs in *Bi. glabrata* which are known to regulate snail immunity[39]. The expansion of *Bu. truncatus* chitinase-like proteins ($n = 103$) is consistent with those in cephalopods[40] and bivalves[41], in which there has been a marked gene expansion relating to the regulation of immune responses[41]. In *Bi. glabrata*, there is evidence that a chitinase-like protein is associated with one or more loci that modulate susceptibility to *S. mansoni*[26].

Also carbohydrate-binding C-type and FREPs/FReD-like lectins could play key roles in the defence of snails against schistosomes[36,42–46]. We identified a significant expansion of genes encoding FReD-like lectins in *Bu. truncatus*, similar to that seen in *Bi. glabrata*. However, the protein members of these species were very distinct, with few homologs clustering together in orthogroups. This finding is consistent with a previous study of *Bu. truncatus*, in which no canonical Class C and/or B IgSF-FREPs were identified[23]. Detailed comparative analysis of FREPs/FReD-like lectins, guided by tertiary structure models, allowed the identification/classification in *Bu. truncatus* of a single IgSF-FREP-like lectin (Class C) and no gene expansion event in this group, in accord with studies of the pond snail, *Physella acuta*[47], the common periwinkle, *Littorina littorea*[48] and the sea hare, *Ap. californica*[49]. Interestingly, there was a marked expansion in the number and diversity of Class A sFReD-like proteins in *Bu. truncatus*, with two large groups with a similar diversification to FREPs in *Bi glabrata*[36]. The presence of additional α-helices and β-sheets in the N-terminal regions of *Bu. truncatus* sFReDs and signal peptide-like domains in some proteins (Class A α) based on tertiary (but not primary) structure models suggest marked functional diversity, which we propose is central to essential immunobiological processes in *Bu. truncatus*. Clearly, further work is needed to establish the roles of sFReDs in *Bu. truncatus* and draw comparisons with information available for *Bi. glabrata*[46], *Ph. acuta*[47], *Mytilus edulis*[50] and other molluscs.

The draft genome (Btru.v1) of *Bu. truncatus* encodes many BIRs/IAPs, calmodulins and Toll-/IL-1-related proteins which have conserved orthologues/paralogues in *Bi. glabrata*. An expansion of BIRs/IAPs was reported earlier for *Bi. glabrata*[12]. While the exact role(s) of these molecules is/are not yet understood, the gastropod and bivalve species studied here encode several copies of genes that might relate to a regulatory role in apoptosis and innate immune response, with an observed gene expansion in snails that act as intermediate hosts for schistosomes[12]. Calmodulins transduce signals in response to increases in intracellular $Ca^{2+}$, represent a major component of calcium-dependent signalling pathways[51] and can play a role in pathogen defence[52]. The diversification of calmodulins in molluscs has been reported previously[53] and has been associated with defence against *S. mansoni* infection in *Bi. glabrata*[27], and against bacteria and yeast[54].

Most invertebrate immune systems include an array of Toll-like receptors (TLRs) that mediate TLR-directed innate immunity to a wide range of pathogen species[55,56]. A much larger number of TLR-like proteins was predicted here from the Btru.v1 genome than reported previously for the transcriptome[23], and more than two-times the number predicted from the *Bi. glabrata* genome[12]. Most of the TLRs predicted for *Bu. truncatus* were categorised (KEGG BRITE) as conserved TLR3/4-like molecules, which are likely to be conserved in invertebrates and mammals[57]. Future work is needed to acurately classify TLRs of planorbid/bulinid snails, to localise their expression in cells and tissues, and to establish which pathogen-associated molecular patterns (PAMPs) they recognise.

The relative conservation of the order of genes in the genomes of the snails studied here indicates that it should be feasible to characterise the genomes and gene orthologues of a range of lymnaeid and physid snails, which are key intermediate hosts (vectors) of socioeconomically important parasitic trematodes other than schistosomes[2] – using the same technological approach as established here. Such an effort could assist in closing some of the knowledge gaps that exist in the understanding of systematics of these groups[22,33], and would provide insight into the molecular evolution of molluscan groups.

The availability of the laboratory (BRI) lines for both *Bu. truncatus* and *S. haematobium* offers excellent opportunities to now study – under well-controlled experimental conditions – the molecular biology of each of these two invertebrates as well as their interactions. *Bu. truncatus* is an essential intermediate host of *S. haematobium*. In an aquatic environment, this snail becomes infected by the miracidium of *S. haematobium*; the ciliated ectoderm of this first larval stage sloughs off upon entry through the snail foot; and the miracidium transforms into a sporocyst, which then undergoes extensive asexual replication within the snail host. Having high-quality genomes and transcriptomes for both *Bu. truncatus* and *S. haematobium* now underpins critical molecular investigations of this asexual phase of reproduction, the cross-talk that occurs between the parasite and the snail host during replication, and the mechanisms and/or processes that govern snail susceptibility and/or immunity to the parasite. We propose that the use of a multi-omics approach[58], involving the use of genome-guided transcriptomic, proteomic, lipidomic and/or glycomic analyses as well as high-resolution single-cell and spatial transcriptomics[59], will strongly complement this focus. In addition, explorations of tertiary structure models for all proteins encoded in snail genomes using *Alpha*Fold[60] should allow the identification of distant orthologues and elucidate dark matter in these proteomes.

In conclusion, defining the first nuclear genome (of ≥ 1 Gb) for a well-defined laboratory line of *Bu. truncatus* opens the door to exploring a range of operational taxonomic units (OTUs) of *Bulinus* from natural populations as well as other key species of snail hosts of schistosomes, as a basis for future systematic, genetic, epidemiological and ecological investigations. Insights into these areas could significantly assist both fundamental and applied studies of schistosomes and schistosomiases, and enable the development of new interventions for this important neglected tropical disease-complex.

## Methods

**Procurement of the snail**. Adult specimens of *Bu. truncatus* originated from a laboratory line (designated BRI), which is routinely maintained in the Biomedical Research Institute (BRI), Rockville, MD, USA[61]. This line was originally sourced from Egypt (Margaret Mentink-Kane, personal communication, 10 October 2020). Individual snails were washed extensively in phosphate-buffered saline (PBS, pH 7.0) and frozen at −80 °C.

**Isolation of high molecular weight genomic DNA, library construction and sequencing**. High quality genomic DNA was isolated from two adult *Bu. truncatus* snails using the Circulomics Tissue Kit (Circulomics, Baltimore, MD, USA). The integrity of the DNA was assessed using Genomic DNA ScreenTape and the Agilent 4200 TapeStation (ThermoFisher, MA, USA). Low molecular weight DNA was removed using a 10 kb short-read eliminator (SRE) kit (Circulomics, Baltimore, MD, USA). The high molecular weight DNA from the two individual snails was used to construct Nanopore Rapid Sequencing (SQK-RAPD004; Oxford Nanopore Technologies) and Ligation Sequencing (SQK-LSK109; Oxford Nanopore Technologies) genomic DNA libraries, according to the manufacturer's instructions. Each flow cell used was washed using a Flow Cell Wash Kit (EXP-WSH003; Oxford Nanopore Technologies, Oxford, UK) and re-used to sequence additional SQK-LSK109 libraries. All libraries were sequenced in the MinION sequencer (Oxford Nanopore Technologies). Following sequencing, bases were converted into FASTQ format from raw FAST5 signals using the program Guppy v.4.2.2 (Oxford Nanopore Technologies). Reads with an average quality (Q) value of <7 were removed. A short-insert (500 bp) genomic DNA library was also constructed using the DNA from one snail, and paired-end sequenced (150 base reads) using TruSeq sequencing chemistry and the NovaSeq sequencing platform

(Illumina, CA, USA). Finally, an in situ Hi-C library was constructed from an additional adult specimen of *Bu. truncatus*, according to manufacturer's instructions (Proximo Hi-C Animal Kit, CA, USA), and paired-end sequenced (150 bp) using the NovaSeq sequencing platform (Illumina).

**Isolation of total RNA, Oxford nanopore library construction and sequencing**. Total RNA that had been isolated from an adult of *Bu. truncatus* (Egyptian strain) in an earlier study[23] was used to prepare a long-read library using the Oxford Nanopore PCR-cDNA Sequencing Kit (SQK-PCS109; Oxford Nanopore Technologies, Oxford, UK), as recommended. This library was sequenced on a MinION sequencer (Oxford Nanopore Technologies) using an EXP-FLP002 flow cell priming kit and a R9.4.1 flow cell (FLO-MIN106). Following sequencing, bases were converted into the FASTQ format from raw FAST5 signals using the program Guppy v. 4.2.2. Reads with an average Q value of <7 were removed.

**Assembly of the genome**. Long reads from the genomic DNA from two *Bu. truncatus* snails were used to assemble contigs using FLYE v2.8-b1674[62] with the --nano-raw option and setting a genome size estimate of 900 Mb. Errors in long read sequence data were initially corrected using medaka_consensus in the medaka package v.1.0.3 (https://github.com/nanoporetech/medaka) and nanopore read data. Contigs polished with the data derived from the short-insert (500 bp) genomic DNA library using pilon v.1.23[63]. Scaffolds were combined with the in situ Hi-C data using 3D-DNA v.180922[64]. Haplotig redundancy was removed by mapping long-read data to the genomic scaffolds, and haplotigs were eliminated employing purge_haplotigs v.1.1.1[65]. Error-corrected long reads were then used to close gaps in scaffolds using TGS-GapCloser v.1.1.1 (https://github.com/BGI-Qingdao/TGS-GapCloser). Following the mapping of short-read data to the genomic scaffolds, haplotig redundancy was eliminated using purge_haplotigs v.1.1.1[65]. The completeness of the genome was assessed (in genome-mode) using BUSCO v 4.0.2[66].

**Assessing genome size, heterozygosity and ploidy**. A short-insert (500 bp) genomic DNA library from a single *Bu. truncatus* snail was used to estimate genome size, heterozygosity and ploidy using the GenomeScope v.2.0 and smudgeplot v.0.2.4 packages[67]. Input into each program was the frequency of 21-mers in the raw short-read data determined using kmc v.3.1.1[68]. Upper and lower frequencies used in smudgeplot were 1300 and 28, respectively. GenomeScope analyses were performed assuming a diploid or tetraploid genome model, according prior evidence for *Bulinus* from the literature[29]. Reads from the short-insert genomic DNA library were also mapped to the reference genome using bwa v.2[69] to estimate minor allelic frequencies and ploidy using PloiPy https://github.com/floutt/PloidPy.

**Predicting repeat-elements, gene models and protein function**. Repeat elements in the genome were predicted using RepeatModeler v. 1.0.8 (http://www.repeatmasker.org) and EDTA v v.1.9.4[70]. Libraries were combined and redundancy was removed using CD-HIT v.4.8.1[71]. The final repeat element library was used to mask the genome using RepeatMasker v4.1.1[72]. Gene models were predicted using funannotate v.1.7.4 (https://github.com/nextgenusfs/funannotate), publicly available RNA-seq data (NCBI BioProject PRJNA680620)[23] (Supplementary Table 1) and inferred protein sequence data sets for *Bi. glabrata* (BB02 strain)[12]. The evidence modeler (EVM) v.1.1.1[73] matrix was weighted as follows: hiq: 7; predicted location of aligned *Bi. glabrata* or Swiss-Prot proteins (accessed 20 December 2020)[25]: 6; PASA v.2.4.1[73]: 5; augustus v.3.3.3[74]: 4; StringTie v2.1.2[75]: 4; and geneMark ES v.3.32[76]: 3. The completeness of the gene set was assessed (in protein-mode) using the tool BUSCO v 4.0.2[66]. The annotation of each inferred amino acid sequence was achieved using InterPro v5.35[77], EggNOG mapper v.5.0[78] and/or homology (E-value ≤ $10^{-5}$) to proteins in the databases Swiss-Prot and TrEMBL within UniProtKB (accessed 20 December 2020)[25], Kyoto Encyclopedia of Genes and Genomes (KEGG)[79] and/or MEROPS release 12[80] using DIAMOND BLASTp v. 0.9.21[81]. Protein groups and pathways were inferred based on homology to KEGG orthology (KO) terms linked to curated KEGG BRITE and pathway hierarchies. Signal peptide domains and/or transmembrane domains were predicted using phobius v.1.04[82]. The sub-cellular localisation of protein sequences was predicted computationally using the program MultiLoc2 v.2.2.25[83]. Evidence of gene transcription was inferred by mapping short and long RNA-seq data to the genome using HISAT2 v.2.1.0[84], and the level of transcription per gene (in transcripts per million, TPM) was inferred using StringTie v2.1.2[75]. Gene models were inferred to have transcriptional support if one or more library had a TPM value of >0.2.

Protein-encoding genes were retained based on the features of their gene models. For each gene, the following features were curated: (1) GeneValidator v.2.1.10[85] score estimated using comparisons to proteins in Swiss-Prot within UniProtKB[25]; (2) evidence of transcription (in TPM); (3) sequence homology (E-value ≤ $10^{-5}$) to proteins in the TrEMBL within UniProtKB[25]; (4) proportion of proteins predicted to be a "low probability subsequence" (LPS) using the program fLPS v.1[86]; (5) GC content for coding domain; (6) length of inferred mRNA sequences; (7) number of exons in the gene model; (8) presence of one or more Pfam conserved domains inferred using InterProScan; and (9) numbers of genes representing individual

groups of orthologous protein between *Bu. truncatus* (Btru.v1) and *Bi. glabrata*[12], established using OrthoFinder v.2.3.11[87]. These features were transformed into normal distributions and subjected to PCA and K-means clustering analyses. Clusters with protein-encoding-like genes were retained for further curation and characterisation. Subsequently, we studied the distribution of repeats in the genome and their association with the 5' and/or 3' untranslated regions (UTRs; 5000 nucleotides for each) of protein-coding genes, employing the Fisher's exact test (p-value < 0.01) to assess statistical significance of association(s).

**Comparative genomic analyses**. Groups of proteins that were orthologous between *Bu. truncatus* (Btru.v1) and *Bi. glabrata* (BB02 strain)[12], *E. chlorotica*[88], *Ap. californica* (https://www.ncbi.nlm.nih.gov/genome/annotation_euk/Aplysia_californica/101/) and/or *P. maximus*[89] (outgroup) were inferred using OrthoFinder v.2.3.11[87] (Table 5). Then, inferred single copy orthologues of Bu. truncatus were mapped to the genomes of *Ac. immaculata* (https://www.ncbi.nlm.nih.gov/assembly/GCA_009760885.1) and *Bi. glabrata* (S1316-R1 strain)[13] using Exonerate v.2.0[90]. For *P. maximus*, the locations of inferred single copy orthologues were identified using respective genome feature format files. Locations of paired single copy orthologues in the same genomic region were grouped using bundlelinks in the program circos v.0.69-8[91] using the following settings: min_bundle_size = 1e4, min_bundle_membership = three or six and max_gap = 1e6. Scaffolds were reordered and displayed using the program circos v.0.69-8[91].

Subsequently, single-copy orthologues were inferred from homologous genes shared by *Bu. truncatus*, *Bi. glabrata*, *E. chlorotica*, *Ap. californica*, *P. maximus* (= outgroup) and/or *Ac. immaculata*, and their amino acid sequences conceptually translated. The clusters of single-copy orthologues representing all five or six species were aligned using the program AQUA[92], employing the programs MUSCLE v3.8.31[93] and MAFFT v.7.271[94] for the alignment and RASCAL v1.34[95] for alignment refinement. Individual clusters of genes with an alignment score of ≥0.8, obtained from the program NorMD[96], were merged using the program PartitionFinder v2.1.1[97] to assign each merged partition to a predicted amino acid substitution matrix. Partitions that did not contain more than 20 amino acids were removed. Remaining partitions were then subjected to separate phylogenetic analyses using the Bayesian inference (BI) and maximum likelihood (ML) tree-building methods. BI analysis was conducted using the program MrBayes v3.2.6[98] from four independent Markov chains, run for 1,000,000 metropolis-coupled MCMC iterations, where trees were sampled every 1000 iterations. The resultant tree was inferred by initially discarding 25% of sampled trees as burn-in, and then using the remaining trees to infer tree topology, branch lengths and to calculate Bayesian posterior probabilities (pp). For ML, a partitioned ML tree was constructed using the program RAxML v8.2.6[99] – selected models for each partition, employing 20 iterations– and the best tree selected for bootstrap analysis (n = 100). A representative tree was prepared using FigTree v.1.31 (http://tree.bio.ed.ac.uk/software/figtree).

**Identification and characterisation of fibrinogen-related domain containing proteins**. Within the *Bu. truncatus* and *Bi. glabrata* protein data sets, proteins with sequence homology to the fibrinogen beta and gamma chains, C-terminal globular domain (Pfam: PF00147.20) were identified using hmmsearch (HMMER v.3.2.1; http://hmmer.janelia.org/) and using a threshold of E-value ≤ $10^{-5}$. Proteins containing a conserved fibrinogen domain were then aligned using hmmalign and using the --trim option and sequences with more than 150 amino acid residues across the conserved fibrinogen domain were retained. Trimmed sequences were de-gapped and re-aligned using MUSCLE v3.8.31[93]. Aligned sequences were then subjected to Bayesian inference (BI) analysis using the program MrBayes v3.2.6[98] and using a WAG model with fixed rate matrices, generating 4,000,000 trees and sampling every 400th tree. The resultant tree was inferred by initially discarding 50% of sampled trees as burn-in, and then using the remaining trees to infer tree topology, branch lengths and to calculate Bayesian posterior probabilities (pp). Phylogenetic trees were rendered and annotated using ggtree (v.1.10.5)[100] in R v.3.4.3 (http://www.R-project.org/). Next, mature peptides (excluding predicted signal peptide domains) were subjected to tertiary structure prediction using AlphaFold[60]. The best ranked model was retained and used in subsequent analyses. FReD-like proteins were classified based on known domains described previously[36], including the presence of a complete or partial C-terminus fibrinogen domain (Class A or A-like, respectively) and with one (Class B; 1-IgSF FREP) or two (Class C) additional N-terminal immunoglobulin superfamily (IgSF) domain. Proteins with FReD domain (Class A) but with additional N-terminal alpha-helices and/or beta-sheets or with complex novel domains were also classified.

**Reporting summary**. Further information on research design is available in the Nature Research Reporting Summary linked to this article.

## Data availability

The nucleotide sequence data linked to the nuclear genome reported in this article is publicly available in the GenBank database and the Sequence Read Archive (SRA; https://www.ncbi.nlm.nih.gov/sra) under the accession numbers SAMN17050146, SAMN16898649 and SAMN16898648 with the NCBI BioProject accession number

PRJNA680620. Protein sequences used for sequence homology searches are available from the Swiss-Prot (UniProtKB; https://www.uniprot.org/help/uniprotkb; accessed 20 December 2020)[25], TrEMBL (UniProtKB; https://www.uniprot.org/help/uniprotkb; accessed 20 December 2020)[25], Encyclopedia of Genes and Genomes (KEGG; https://www.genome.jp/kegg/; accessed 20 December 2020)[79] and MEROPS release 12 (https://www.ebi.ac.uk/merops/)[80] databases. All other data used are referred to in this article and its supplementary files.

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

## Acknowledgements

This research project was supported by grants from the Australian Research Council (ARC) to R.B.G., N.D.Y. and P.K.K. Informatics was supported via the LIEF HPC-GPGPU facility at the University of Melbourne, with assistance from LIEF grant LE170100200. The following was provided by the NIAID Schistosomiasis Resource Center for distribution through BEI Resources, NIAID, NIH: *Bulinus truncatus* subsp. *truncatus*, NR-21971.

## Author contributions

N.D.Y. conceived, planned and developed the study, conducted the laboratory work, developed methods, wrote the original draft and funded the project. A.J.S. assisted with methods, laboratory work and edited the manuscript. T.W. assisted with laboratory work and edited the manuscript. P.K.K. assisted with informatics, funding and commented on the manuscript. M.M-K. reared and provided the snails used in this study. J.R.S. and D.R. both reviewed and edited the manuscript. R.B.G. was involved in planning and developing the conceptual framework, co-wrote the original draft and funded the project. All authors read, commented on and approved the submitted version of the manuscript.

## Competing interests

The authors declare no competing interests.
