## [Peer Review File · Nature Communications]

Nuclear genome of *Bulinus truncatus*, an intermediate host of the carcinogenic human blood fluke *Schistosoma haematobium*Reviewers' Comments:

Reviewer #1:

Remarks to the Author:

The manuscript by Young et al describes the nuclear genome of a snail *Bulinus truncatus*. The species is a host to a pathogen *Schistosoma haematobium*.

Overall, while the genome is an important piece of the puzzle, the analyses shown are relatively general. The term "advanced sequence-informatics" in the abstract is rather confusing. The paper itself is very brief and, to my eyes, does not reveal anything outstanding about this host.

Some analyses seem to come out-of-nowhere - e.g., "synteny", and it is unclear what the message here is. Some figures (Figure 2) are rather of technical nature (should be moved to the supplement?). Why didn't authors look more extensively at the repetitive elements? I also wonder why authors did not explore genomic regions around the expanded gene families, which would have given some more insights into possible adaptation signatures.

Technically, it should have been possible to achieve chromosomal-scale assembly. It would certainly have helped with some of the questions and comparisons. The discussion on this point is unclear/missing. Their HiC map in Suppl Fig 1 looks cropped out (and chromosomal scaffolds/units not visible)?

The paper as-such would thus rather be suited to be published as a genome resource in another journal. However, I would welcome any attempt by the authors to include more analyses to make the novel manuscript findings stand out and have a better flow of the findings.

Reviewer #2:

Remarks to the Author:

This report provides valuable data from a first analysis of the genome of the snail *Bulinus truncatus*, snail vector of several species of flatworm parasites including *Schistosoma haematobium*, cause of urinary schistosomiasis, a major human infectious disease that may induce bladder cancer. The availability of a *Bu. truncatus* genome allows for "omics"-level comparisons with other Mollusca (lophotrochozoan protostome invertebrates), a continually understudied phylogenetic clade among Animalia. Moreover, important genomic analyses can now be made with a most distantly related snail species within the family Planorbidae, *Biomphalaria glabrata*, specific vector of another schistosome species, *Schistosoma mansoni*. This may reveal traits that define snail vector suitability/compatibility and parasite virulence, of great relevance toward study of epidemiology and future control efforts aimed at schistosomiasis as infectious diseases.

The work is performed using well defined samples from *Bulinus truncatus*, confirmed vector host of *Schistosoma haematobium*, originally from Egypt but maintained long term in the USA under NIH resource contract at BEI. The same international expert investigative team has recently also contributed complementary studies reporting the mitogenome and transcriptomic analyses for this snail strain. The genome assembly was generated using modern and appropriate sequencing techniques and bioinformatics approaches, with data publicly available. Analyses performed logically address relevant aspects of molluscan genome biology (repetitive content, synteny, orthology, gene expansions) and of factors indicated by previous work with *Bi. glabrata*/*S.mansoni* to have a role in snail vector/parasite interactions.

Comments.

-Valid comparisons (including synteny, orthologs, antiparasite factors) are made with Mollusca for which relevant genome assemblies are available. Implications of findings can be confusing, however, due to complex phylogenetic relationships. Some molluscs represent (remote) different classes

(bivalves versus gastropods) whereas other belong to the same class, group (Lymnaeoidea, L299,344) or even family; Achatina (land snails, stylommatophora) versus Bulinus and Biomphalaria (planorbidae within the Hygrophylid freshwater snails). Clarification of the close relation between the latter two as planorbid vectors for distinct schistosomes will further underscore the significance of this study. This issue can be addressed by including a figure showing a tree of the basic phylogeny of molluscs included in the analyses.

-Regarding analyses of anti-parasite factors in *Bu truncatus*,

a) Given that the genes of GRC and PTC2 regions do not confer actual resistance to *Bi glabrata* for *S. mansoni*, but rather associate with reduced yet persistent production of cercaria following infection, it is recommended to refer to these complexes as (L292) as associated with "reduced susceptibility" rather than "resistance" (the genes in the regions do NOT prevent parasite exposure from developing patent, productive infections). Likewise L310-1, recommend ".....one or more loci that modulate susceptibility to *S. mansoni*" (or similar wording)

b) Regarding FREPs/FReDs and lectins L312, note that FREPs/FreDs are actually categories of lectins, just like the C-type lectins. In L313, also cite the report by Li et al, 2020 Coordination of humoral immune factors dictates compatibility between *Schistosoma mansoni* and *Biomphalaria glabrata*, eLife 2020;9:e51708 doi: 10.7554/eLife.51708 in support of a possible role of FREPs in antiparasite defense. L317-9 consider deleting/modifying this section; the study cited did not attempt to detect IgSF sequences, only PCR targeting the fibrinogen domain. L319- restate to (wording similar to)Absent IgSF domains these proteins in *Bu. truncatus* are not FREPs but qualify as FreDs. The statement "IgSF domain is..... essential for functionality of FRED-like proteins as hemolymph lectins" is incorrect. Note that FreD lectins (containing FBG but not IgSF) also function in molluscan defenses. Consider the following information/references:

"Similarly, a group of FreD-containing proteins (ngiopoietin-4, ficolin-2) that lacked N-terminal Ig-fold(s) were identified as a distinct group of FREP-like proteins, separate from the VigL lectin family." Wu XJ, Dinguirard N, Sabat G, Lui HD, Gonzalez L, Gehring M, Bickham-Wright U, Yoshino TP. Proteomic analysis of *Biomphalaria glabrata* plasma proteins with binding affinity to those expressed by early developing larval *Schistosoma mansoni*. PLoS Pathog. 2017 May 16;13(5):e1006081. doi: 10.1371/journal.ppat.1006081. PMID: 28520808; PMCID: PMC5433772.

"The *Physella acuta* reference transcriptome also revealed 24 unique transcripts encoding proteins consisting of a single fibrinogen-related domain (FreDs), with a short N-terminal sequence encoding either a signal peptide, transmembrane domain or no predicted features. The *Physella acuta* FreDs are candidate immune genes..." Schultz JH, Bu L, Adema CM. Comparative immunological study of the snail *Physella acuta* (Hygrophila, Pulmonata) reveals shared and unique aspects of gastropod immunobiology. Mol Immunol. 2018 Sep;101:108-119. doi: 10.1016/j.molimm.2018.05.029. Epub 2018 Jun 17. PMID: 29920433.

"In molluscs haemolymph lectins bearing fibrinogen-like domain (FREP) act as immune pattern-recognition receptors. A full-length cDNAs of MytFREP1 and MytFREP2 cloned from haemocytes of blue mussel *Mytilus edulis* encoded putative polypeptides of 230 and 241 amino acids. Both polypeptides consist of signal peptide and C-terminal fibrinogen-like domain." Gorbushin AM, Iakovleva NV. A new gene family of single fibrinogen domain lectins in *Mytilus*. Fish Shellfish Immunol. 2011 Jan;30(1):434-8. doi: 10.1016/j.fsi.2010.10.002. Epub 2010 Oct 15. PMID: 20951811.

Minor comments.

-Abstract, and L53-54 it is unconventional to contrast free-living snails with snails that act as vectors.

-L63 "entry or exit" of the parasite?

-L52-88, include the word "vector" for snail intermediate host at least once. L64?

-L85-88 and also perhaps in the discussion, suggest to include the concept of comparative snail vector "omics", capitalizing on the availability of the *Biomphalaria* and *Bulinus* genomes/transcriptomes (and possibly other snail vectors like Lymnaeid and Physid snails).

-Gene annotation yielded 26,292 protein coding genes that were annotated(L120) and 21951 genes that were functionally annotated (L142), explain/reconcile the difference.

-This reviewer asks for clarification of interpretation of the results for the investigation of (origin of?) ploidy. *Bulinus truncatus* from the field was reported twice to be tetraploid, please consider additional reference showing karyogram (Yaseem 199 Cytogenetics and biology of the intermediate host of human bilharziasis, *Bulinus truncatus* common in upper Egypt, *Cytologia* 58, 53-60), and fittingly the snail is indicated to be tetraploid (L274). The current statements that (L105) "data matched a diploid model" and (L280-1) "polymorphism in the genome (.....) was consistent with that of diploid organism" deserves additional explanation/interpretation of why this is "supporting autotetraploidy with limited chromosomal divergence".

Also, with the notion of whole genome duplication (WGD) proposed for *Achatina* species, do similar biological processes underlie the ploidy variations in the genus *Bulinus*? (Liu C, Ren Y, Li Z, Hu Q, Yin L, Wang H, Qiao X, Zhang Y, Xing L, Xi Y, Jiang F, Wang S, Huang C, Liu B, Liu H, Wan F, Qian W, Fan W. Giant African snail genomes provide insights into molluscan whole-genome duplication and aquatic-terrestrial transition. *Mol Ecol Resour.* 2021 Feb;21(2):478-494. doi: 10.1111/1755-0998.13261. Epub 2020 Oct 21. PMID: 33000522.)

-Hi-C sequencing was applied toward construction of large scaffolds and synteny analyses. This technique can enable chromosome level assemblies, but this was apparently not achieved for *Bu truncatus*. Please clarify whether or not fig S1 provides some level of representation of chromosomal units. Regardless, the authors should discuss in the manuscript why chromosome level assembly was not achieved. If this is due to the tetraploidy or because of (technical challenges due to)the repetitive nature of the genome, will that also challenge assembly of other gastropods? Such information is valuable to avoid unrealistic expectations that may lead to undervaluing this and future gastropod genome characterizations. This gains importance in light of the suggestion to apply similar approaches to additional (species of) vector snails(L359-361)

REVIEWER #1:

1.1. “The term "advanced sequence-informatics" in the abstract is rather confusing. The paper itself is very brief and, to my eyes, does not reveal anything outstanding about this host.”

1.1. RESPONSE: We have clarified and enhanced the abstract (Lines 14 – 17). We agree that the paper is brief, but we disagree that it “does not reveal anything outstanding. Many *Bulinus* species are important vectors for schistosomiasis. The abstract has been modified to include: “Here, we define the first genome for a key intermediate host of *S. haematobium* – called *Bulinus truncatus* – and explore key protein groups inferred to play an integral role in the snail’s biology and its relationship with the schistosome parasite. This work now solidly underpins fundamental studies of snail-parasite interactions in the search for targets to block schistosomiasis transmission.”

1.2 “Some analyses seem to come out-of-nowhere - e.g., "synteny", and it is unclear what the message here is.”

1.2. RESPONSE: In the revised manuscript, we have enhanced the narrative and flow for the reader (see Lines 184 to 189 and 204 to 209).

1.3. “Some figures (Figure 2) are rather of technical nature (should be moved to the supplement?).”

1.3. RESPONSE: We agree. We have elected to move Fig. 2 to Supplementary Material (now Supplementary Fig. 5).

1.4. “Why didn't authors look more extensively at the repetitive elements? I also wonder why authors did not explore genomic regions around the expanded gene families, which would have given some more insights into possible adaptation signatures.”

1.4. RESPONSE: Thank you for this question. We have now extended the exploration of repetitive elements and described this in the revised manuscript (lines 108 to 127; see also Supplementary Tables 3 and 4 and Supplementary Fig. 3.

1.5. “Technically, it should have been possible to achieve chromosomal-scale assembly. It would certainly have helped with some of the questions and comparisons. The discussion on this point is unclear/missing. Their HiC map in Suppl Fig 1 looks cropped out (and chromosomal scaffolds/units not visible)?”

1.5. RESPONSE: The *Bulinus truncatus* HiC data was crucial to have achieved scaffold lengths reported here. The main limitation was the complexity associated with tetraploidy and the use of a single enzyme in the commercial kit (Phase Genomics). We now discuss these aspects better and which steps will be needed to ‘polish’ the genome (lines 350-354). Supplementary Fig.1 was not cropped. We elected to include the contact map among all scaffolds; e now show these contacts with and without scaffold start and end points for clarity.

1.6. “The paper as-such would thus rather be suited to be published as a genome resource in another journal. However, I would welcome any attempt by the authors to include more analyses to make the novel manuscript findings stand out and have a better flow of the findings.”

1.6. RESPONSE: Thank you for this comment. While the manuscript is rather technical, it does represent a major breakthrough for a critically important host snail. To address, this critique, we have undertaken a number of additional analyses to make the findings stand out and have enhanced the flow of the findings. We now summarise key protein families (Tables 5-8 and Figs. 3-5), explore repeat elements (Supplementary Fig. 3 and Supplementary Tables 3 and 4) and focus on a detailed characterisation of novel lectins found in planorbid snails (Fig. 4 and 5, Supplementary Table 8 and Supplementary Fig. 7). We hope that these improvements alleviate the reviewer’s concern.

REVIEWER #2:

2.1. “Implications of findings can be confusing, however, due to complex phylogenetic relationships. Some molluscs represent (remote) different classes (bivalves versus gastropods) whereas other belong to the same class, group (Lymnaeidae, L299,344) or even family; Achatina (land snails, stylommatophora) versus *Bulinus* and *Biomphalaria* (planorbidae within the Hygrophylid freshwater snails). Clarification of the close relation between the latter two as planorbid vectors for distinct schistosomes will further underscore the significance of this study. This issue can be addressed by including a figure showing a tree of the basic phylogeny of molluscs included in the analyses.”

2.1. RESPONSE: We are grateful for this suggestion. We have now included a phylogenetic tree summarising the relationships of the molluscan species included (Supplementary Fig. 6) and ensure that their taxonomy is clarified in the text (see lines 184-189, 204-209 and 335 to 336).

2.2. “Regarding analyses of anti-parasite factors in *Bu truncatus*: a) Given that the genes of GRC and PTC2 regions do not confer actual resistance to *Bi glabrata* for *S. mansoni*, but rather associate with reduced yet persistent production of cercaria following infection, it is recommended to refer to these complexes as (L292) as associated with “reduced susceptibility” rather than “resistance” (the genes in the regions do NOT prevent parasite exposure from developing patent, productive infections). Likewise L310-1, recommend “.....one or more loci that modulate susceptibility to *S. mansoni*” (or similar wording)”

2.2. RESPONSE: Thank you for raising this issue. We have amended the text to refer to “reduced susceptibility” and have refrained from using the term resistance throughout. We have also modified the text to read (line 379 to 380): “...one or more loci that modulate susceptibility to *S. mansoni*.”

2.3. “Regarding analyses of anti-parasite factors in *Bu truncatus*: b) Regarding FREPs/FReDs and lectins L312, note that FREPs/FReDs are actually categories of lectins, just like the C-type lectins. “

2.3. RESPONSE: We have clarified this issue in the text (line 381) and have further categorised/characterised sFReD-like proteins of *Bu. truncatus* and critically assessed their relationship to known sFReD, FREP and FREM proteins of *Bi. glabrata*. These results are described (lines 283-328) and discussed (lines 381 -396) in the revised text.

2.4. “Regarding analyses of anti-parasite factors in *Bu truncatus*: b) In L313, also cite the report by Li et al, 2020 Coordination of humoral immune factors dictates compatibility between *Schistosoma mansoni* and *Biomphalaria glabrata*, eLife 2020;9:e51708 doi: 10.7554/eLife.51708 in support of a possible role of FREPs in antiparasite defense.”

2.4. RESPONSE: Thank you for raising this point. We have added this reference to the text and discussed their findings (lines 381-382).

2.5. “Regarding analyses of anti-parasite factors in *Bu truncatus*: b) L317-9 consider deleting/modifying this section; the study cited did not attempt to detect IgSF sequences, only PCR targeting the fibrinogen domain.

2.5. RESPONSE: We have deleted this text as suggested. Due to the lack of published work on *Bulinus* FReD-like lectins and their differences from FREP/sFReD/FREM lectins of *Bi. glabrata*, we undertook a further, detailed analyses of these novel lectins of *Bulinus truncatus* (Fig. 5, Supplementary Table 8, Supplementary Fig. 7 and Supplementary File 1). Using structural predictions (employing the most advanced machine-learning based modelling algorithm), we searched again for IgSF domains and identified one possible candidate that was not detected using primary sequence-based homology. These findings have been described and discussed (also with reference to works suggested by reviewer 2).

2.6. “Regarding analyses of anti-parasite factors in *Bu truncatus*: b) L319- restate to (wording similar to)Absent IgSF domains these proteins in *Bu. truncatus* are not FREPs but qualify as FReDs. The statement “IgSF domain is..... essential for functionality of FRED-like proteins as hemolymph lectins” is incorrect. Note that FReD lectins (containing FBG but not IgSF) also function in molluscan defenses. Consider the following information/references:

“Similarly, a group of FReD-containing proteins (ngiopoietin-4, ficolin-2) that lacked N-terminal Ig-fold(s) were identified as a distinct group of FREP-like proteins, separate from the VigL lectin family.” Wu XJ, Dinguirard N, Sabat G, Lui HD, Gonzalez L, Gehring M, Bickham-Wright U, Yoshino TP. Proteomic analysis of *Biomphalaria glabrata* plasma proteins with binding affinity to those expressed by early developing larval *Schistosoma mansoni*. PloS Pathog. 2017 May 16;13(5):e1006081. doi: 10.1371/journal.ppat.1006081. PMID: 28520808; PMIDID: PMC5433772.

“The *Physella acuta* reference transcriptome also revealed 24 unique transcripts encoding proteins consisting of a single fibrinogen-related domain (FReDs), with a short N-terminal sequence encoding either a signal peptide, transmembrane domain or no predicted features. The *Physella acuta* FReDs are candidate immune genes...”

Schultz JH, Bu L, Adema CM. Comparative immunological study of the snail *Physella acuta* (Hydrophila, Pulmonata) reveals shared and unique aspects of gastropod immunobiology. Mol Immunol. 2018 Sep;101:108-119. doi: 10.1016/j.molimm.2018.05.029. Epub 2018 Jun 17. PMID: 29920433.

“In molluscs haemolymph lectins bearing fibrinogen-like domain (FREP) act as immune pattern-recognition receptors. A full-length cDNAs of MytFREP1 and MytFREP2 cloned from haemocytes of blue mussel *Mytilus edulis* encoded putative polypeptides of 230 and 241 amino acids. Both polypeptides consist of signal peptide and C-terminal fibrinogen-like domain.” Gorbushin AM, Iakovleva NV. A new gene family of single fibrinogen domain lectins in *Mytilus*. Fish Shellfish Immunol. 2011 Jan;30(1):434-8. doi: 10.1016/j.fsi.2010.10.002. Epub 2010 Oct 15. PMID: 20951811.

2.6. RESPONSE: We are grateful for this critique. We have now amended and enhanced the text accordingly (see also RESPONSES to 2.3 and 2.5).

2.7. “-Abstract, and L53-54 it is unconventional to contrast free-living snails with snails that act as vectors.”

2.7. RESPONSE: As complete genomes are not available for most snails that act as vectors, we elected to use additional snail species for comparative analyses. Thus, we believe that this is justified, which is why we elect to retain ‘free-living snails’.

2.8. “-L63 “entry or exit” of the parasite? “

2.8. RESPONSE: We have addressed this issue in the text (line 63)

2.9. “-L52-88, include the word “vector” for snail intermediate host at least once. L64?”

2.9. RESPONSE: We have addressed this issue in the text (line 54)

2.10. -L85-88 and also perhaps in the discussion, suggest to include the concept of comparative snail vector “omics”, capitalizing on the availability of the Biomphalaria and Bulinus genomes/transcriptomes (and possibly other snail vectors like Lymnaeid and Physid snails).”

2.10. RESPONSE: We have amended the text accordingly (lines 87-89; lines 417 to 418).

2.11. “-Gene annotation yielded 26,292 protein coding genes that were annotated(L120) and 21951 genes that were functionally annotated (L142), explain/reconcile the difference.”

2.11. RESPONSE: Many genes had homologues in related taxa (e.g., *Bi. glabrata*) but they could not be functionally annotated with GO terms or assigned to known protein families or biochemical or biological pathways. This is a commonly encountered issue for ‘non-model’ organisms.

2.12.” -This reviewer asks for clarification of interpretation of the results for the investigation of (origin of?) ploidy. *Bulinus truncatus* from the field was reported twice to be tetraploid, please consider additional reference showing karyogram (Yaseem 199 Cytogenetics and biology of the intermediate host of human bilharziasis, *Bulinus truncatus* common in upper Egypt, *Cytologia* 58, 53-60), and fittingly the snail is indicated to be tetraploid (L274). The current statements that (L105) “data matched a diploid model” and (L280-1) “polymorphism in the genome (.....) was consistent with that of diploid organism” deserves additional explanation/interpretation of why this is “supporting autotetraploidy with limited chromosomal divergence”

2.12. REPOSENSE: We agree that this snail is tetraploid, but the genomic data did not fit a tetraploid model, as stated in the text (lines 100-106). This finding indicates recent autotetraploidy, as discussed (lines 341-354). This aspect has been discussed further in the text (lines 341-354). We have also added the additional reference suggested by this reviewer.

2.13. “Also, with the notion of whole genome duplication (WGD) proposed for *Achatina* species, do similar biological processes underlie the ploidy variations in the genus *Bulinus*? (Liu C, Ren Y, Li Z, Hu Q, Yin L, Wang H, Qiao X, Zhang Y, Xing L, Xi Y, Jiang F, Wang S, Huang C, Liu B, Liu H, Wan F, Qian W, Fan W. Giant African snail genomes provide insights into molluscan whole-genome duplication and aquatic-terrestrial transition. *Mol Ecol Resour.* 2021 Feb;21(2):478-494. doi: 10.1111/1755-0998.13261. Epub 2020 Oct 21. PMID: 33000522.)”

2.13. REPOSENSE: A critical appraisal of the supplementary material of the manuscript revealed genetic polymorphisms consistent with a whole-genome duplication event. As this further supports the possibility of tetraploidy arising in *Bu. truncatus* due to auto-tetraploidy, we included this aspect and key references, enhancing in our discussion (lines 348-349).

2.14. “-Hi-C sequencing was applied toward construction of large scaffolds and synteny analyses. This technique can enable chromosome level assemblies, but this was apparently not achieved for *Bu truncatus*. Please clarify whether or not fig S1 provides some level of representation of chromosomal units. Regardless, the authors should discuss in the manuscript why chromosome level assembly was not achieved. If this is due to the tetraploidy or because of (technical challenges due to) the repetitive nature of the genome, will that also challenge assembly of other gastropods? Such information is valuable to avoid unrealistic expectations that may lead to undervaluing this and future gastropod genome characterizations. This gains importance in light of the suggestion to apply similar approaches to additional (species of) vector snails(L359-361)”

2.15. RESPONSE: This point was addressed above (see 1.5 RESPONSE – reviewer 1).

CONCLUDING REMARKS

We are grateful for the constructive criticisms and suggestions. We have addressed all of the issues and have modified the manuscript accordingly (with changes highlighted in yellow). We believe that the R1

manuscript now meets the high standard required for publication in *Nature Communications*. We look forward to the final decision on our manuscript.

Reviewers' Comments:

Reviewer #1:

Remarks to the Author:

We appreciate that the authors have attempted to revise and improve the manuscript. While it is more sound now, we still find the findings too technical and preliminary to make any substantial impact on the field. The main novelty, to us, seems to be located in the sections on "Protein groups inferred to be involved in the snail-schistosome relationship" onwards, which seem to rely mainly on previously published data and the "usual suspects" among expanding or other gene families. It is lacking a story or some testable hypotheses, it is sometimes unclear why authors looked at a particular gene family and how unique it is to the system. More could have been done, also in light of the available laboratory lines. Genome sequence (accompanied by more in-depth analyses) would have allowed to understand if any of the reported gene families are particularly fast evolving not just in coding sequence but maybe in their regulatory regions or genome placement, their expression could have been studied (during different states of infection to show any difference or putative functional implication). Etc. We also find some of the technical conclusions still lacking validation, for example the tetraploid nature of the genome should be verified and visible in Figure 2 synteny plots (and it is not, at least to us) as well as gene family expansion analyses (e.g., Fig 5). In summary, while we completely agree that the system and the set of questions is very important, the report of the genome sequence per-se - at this quality and with the set of basic analyses - does not shed much light on the genome "biology" as such.

Reviewer #2:

Remarks to the Author:

The revisions effected by the authors have further increased the value of the manuscript. Minor issues remain to be clarified/corrected.

Response to reviewer comment 2.14 (and 1.5)

Lines 350-354, or elsewhere do not seem to discuss that the limitation for chromosome-scale assembly was "the complexity associated with tetraploidy and the use of a single enzyme in the commercial kit (Phase Genomics)."

Please include brief statement to this effect in response to reviewer comment 2.14 "Such information is valuable to avoid unrealistic expectations that may lead to undervaluing this and future gastropod genome characterizations" Possibly following L339 for the revision" ...combination of second- (short-read) and third-generation (long-read) sequence data...."

Re response to 2.7

The term "free-living" is still problematic. It is not effective to differentiate vector snails from non-vector snails. Non-vector snails may not exist; virtually all snail species are host in at least one parasite life cycle. For instance, *Achatina* (intended to represent free-living snails?) transmit *Angiostrongylus* nematodes. Parasitology terminology refers to free-living for parasite stages in the environment, after emergence from the host, before re-entry into the next host. Free-living is also applied to non-parasitic organisms. Free-living does not apply to (host)snails. Vector snails that exist in the environment are also free-living.

Rewording is highly recommended to avoid ambiguity.

Line 346 correct "Bu. truncatus is proposed to have four chromosomes (tetraploid) 29,30"

Figure 5: To facilitate location of the *Bulinus* 2-IgSF FREP-like (Btru_028592-T1), designate with a light blue (*Bulinus*) square (2 IgSF_FREP)? Also explain/correct how for this sequence the code Btru_028592-T1 (line 327, and

gene tree figure 5) relates to Btru_007887-T1 (bottom fig5B, Sup fig 7)

Supplementary tables: the excel file [Supplementary Tables Supplementary Dataset (7921KB) Source File (XLSX) 7921KB]does not include tables 6,7,8? Please update.

Reviewer #3:

None

REVIEWER #1:

1.1. “we still find the findings too technical and preliminary to make any substantial impact on the field.”

***Response:** We respectfully disagree. The manuscript describes findings that are much more than technical and preliminary, and that provide a very valuable, publicly accessible resource for many researchers working on snails and/or on pathogens that they carry or transmit. Areas include taxonomy, population genetics/genomics, evolutionary biology, ecology and epidemiology and host-pathogen interactions.

1.2. “It is lacking a story or some testable hypotheses, it is sometimes unclear why authors looked at a particular gene family and how unique it is to the system. “More could have been done, also in light of the available laboratory lines.”

***Response:** These statements are subjective and unconstructive. If nothing is known about a topic, sometimes one needs to start somewhere. The story is clearly described and findings cautiously interpreted. Our analysis of the genome and key molecular groups are particularly important for exploring snail–schistosome interactions/compatibility and demonstrate the utility of this new genomic resource. The selection of a particular gene family relates to the biological importance and questions surrounding this gene family and the interest from experts working in this field (including the reviewers who requested detailed insight into this family, upon first revision of the manuscript). Clearly, the present study, which was far from a trivial exercise, is critically important and paves the way for many hypotheses to be tested. It overcomes major limitations associated with the dearth of molecular information for members of the genus *Bulinus*, and provides a solid foundation for work on a very well-defined laboratory strain of *B. truncatus* which is readily provided (at no cost) via the Biomedical Research Institute (BRI) in the USA to scientists working in any country around the world. More can always be done, and the present genome provides researchers worldwide with the resources that they need to collectively accelerate research in this exciting field.

1.3. “Genome sequence (accompanied by more in-depth analyses) would have allowed to understand if any of the reported gene families are particularly fast evolving not just in coding sequence but maybe in their regulatory regions or genome placement, their expression could have been studied (during different states of infection to show any difference or putative functional implication).”

***Response:** The aim of the present study was clear and well-defined (lines 58-62). Our goal was to provide a major step forward for the field, and we have achieved this (cf. previous points). While the topics that the reviewer raises are relevant to the parasitological community, they are beyond the aims of the present study.

1.4. “We also find some of the technical conclusions still lacking validation, for example the tetraploid nature of the genome should be verified and visible in Figure 2 synteny plots (and it is not, at least to us) as well as gene family expansion analyses (e.g., Fig 5).”

***Response:** *Bulinus truncatus* is known to be tetraploid; the k-mer analyses indicate clearly that the sequence data do not strictly fit a diploid model. As we did not achieve complete chromosome contiguity, we elected to present a single copy of each scaffold with no inference of haplotigs. This is presented in Fig. 2 (now Fig. 1) and includes the gene models presented in Fig. 5 (now Fig. 4). Our findings are clearly and transparently presented, and carefully interpreted; the findings will be validated by the community over time, as is the case for most genome studies.

1.5. “In summary, while we completely agree that the system and the set of questions is very important, the report of the genome sequence per-se - at this quality and with the set of basic analyses - does not shed much light on the genome "biology" as such.”

***Response:** We thank the reviewer for acknowledging the importance of ‘the system and the set of questions’. This acknowledgement somewhat contradicts a previous suggestion that we ‘lack a story’ (cf. 1.2). The evidence presented by us shows that we have shed light on some key aspects of the biology of *B. truncatus* and that we have produced critical resources to the scientific

community. Through this work, we advance the field of snail genomics and, clearly, enable future progress in this area.

REVIEWER #2:

2.1. Response to reviewer comment 2.14 (and 1.5) Lines 350-354, or elsewhere do not seem to discuss that the limitation for chromosome-scale assembly was “the complexity associated with tetraploidy and the use of a single enzyme in the commercial kit (Phase Genomics).” Please include brief statement to this effect in response to reviewer comment 2.14 “Such information is valuable to avoid unrealistic expectations that may lead to undervaluing this and future gastropod genome characterizations” Possibly following L339 for the revision” ...combination of second- (short-read) and third-generation (long-read) sequence data...”

Response: We have enhanced the statement (lines 327 to 329) to now state: “Limited genetic divergence among the four sets of chromosomes and/or the use of a single restriction enzyme (*DpnII*) for Hi-C library construction, are probable reasons for the *Bu. truncatus* scaffolds being shorter than the expected chromosome lengths.”

2.2. Re: response to 2.7: The term “free-living” is still problematic. It is not effective to differentiate vector snails from non-vector snails. Non-vector snails may not exist; virtually all snail species are host in at least one parasite life cycle. For instance, *Achatina* (intended to represent free-living snails?) transmit *Angiostrongylus* nematodes. Parasitology terminology refers to free-living for parasite stages in the environment, after emergence from the host, before re-entry into the next host. Free-living is also applied to non-parasitic organisms. Free-living does not apply to (host)snails. Vector snails that exist in the environment are also free-living. Rewording is highly recommended to avoid ambiguity.

***Response:** We agree with this point and address this in the revised version of the manuscript (lines 31 and 32). We simply state that many snails can act as intermediate hosts (i.e. vectors) of parasites of vertebrates including humans.

2.3. Line 346 correct “*Bu. truncatus* is proposed to have four chromosomes (tetraploid) 29,30”

***Response:** We thank the reviewer for identifying this error. We now state: “*Bu. truncatus* is proposed to have four sets of chromosomes (tetraploid)”

2.4. Figure 5: To facilitate location of the *Bulinus* 2-IgSF FREP-like (*Btru_028592-T1*), designate with a light blue (*Bulinus*) square (2 IgSF_FREP)? Also explain/correct how for this sequence the code *Btru_028592-T1* (line 327, and gene tree figure 5) relates to *Btru_007887-T1* (bottom fig5B, Sup fig 7)

***Response:** We thank the reviewer for identifying this error. Indeed, the accession number in the body of the text was incorrect. The text (lines 299 to 301) now states: “Interestingly, the tertiary structure of a Class C-like *Bu. truncatus* protein (*Btru_007887*) in this group encoded two IgSF-like domains (cf. Fig. 4b)”. This accession number is now correct and consistent with that presented in Fig. 5b (now Fig. 4b).

2.5. Supplementary tables: the excel file [Supplementary Tables Supplementary Dataset (7921KB) Source File (XLSX) 7921KB]does not include tables 6,7,8? Please update.

***Response:** We thank the reviewer for identifying this error. All supplementary tables are now included in the revised (R2) version of the manuscript. To address editorial changes, supplementary tables are now in the PDF format, except for Tables 4 and 6, which are now included as Supplementary Data files 1 and 2, respectively.

Concluding remarks: We are grateful for the additional reports, comments and suggestions. We believe that the newly revised (R2) manuscript has been further enhanced and now meets the high standard required for publication in *Nature Communications*. We look forward to the final decision on our manuscript.